# Route Before Retrieve: Activating Latent Routing Abilities of LLMs for RAG vs. Long-Context Selection

## Abstract

Recent advances in large language models (LLMs) have expanded the context window to beyond 128K tokens, enabling long-document understanding and multi-source reasoning. A key challenge, however, lies in choosing between **retrieval-augmented generation (RAG)** and **long-context (LC)** strategies: RAG is efficient but constrained by retrieval quality, while LC supports global reasoning at higher cost and with position sensitivity. Existing methods such as *Self-Route* adopt failure-driven fallback from RAG to LC, but remain passive, inefficient, and hard to interpret. We propose **Pre-Route**, a proactive routing framework that performs structured reasoning *before* answering. Using lightweight metadata (e.g., document type, length, initial snippet), Pre-Route enables task analysis, coverage estimation, and information-need prediction, producing explainable and cost-efficient routing decisions. Our study shows three key findings: (i) LLMs possess latent routing ability that can be reliably activated with guidelines, allowing single-sample performance to approach that of multi-sample (Best-of-N) results; (ii) linear probes reveal that structured prompts sharpen the separability of the "optimal routing dimension" in representation space; and (iii) distillation transfers this reasoning structure to smaller models for lightweight deployment. Experiments on LaRA (in-domain) and LongBench-v2 (OOD) confirm that Pre-Route outperforms Always-RAG, Always-LC, and Self-Route baselines, achieving superior overall cost-effectiveness.

## 1 Introduction

In recent years, Large Language Models (LLMs) have shown unprecedented capabilities in processing long contexts. The latest models—such as GPT-5, DeepSeek R1, and Qwen3—support inputs beyond 128K tokens (OpenAI, 2025; DeepSeek, 2025; Yang et al., 2025), enabling tasks like long-document comprehension, cross-segment reasoning, and multi-source synthesis. Yet, how to efficiently and accurately exploit these capabilities across tasks remains an open challenge (Liu et al., 2024). Two mainstream approaches have emerged: **Retrieval-Augmented Generation (RAG)**, which retrieves relevant snippets from external sources (Lewis et al., 2020), and **Long Context (LC)** processing, where the full context is provided to the model for end-to-end reasoning (Kamradt, 2023; Hsieh et al., 2024).

These two pathways differ in suitability, each with distinct strengths and limitations. Benchmarks (Li et al., 2025; Bai et al., 2024a; An et al., 2024) show that LC excels at comparison and complex reasoning by leveraging a global view, while RAG is preferred for fact retrieval and hallucination-sensitive tasks, offering precise evidence and abstention. LC works well for well-structured texts (e.g., financial reports) but suffers from positional effects such as lost-in-the-middle problems (Liu et al., 2024). Conversely, RAG adapts better to loosely structured texts (e.g., novels) and is less position-sensitive, though its effectiveness depends on retrieval quality. Thus, choosing between RAG and LC is a key challenge for advancing long-context processing.

Existing strategies such as Self-Route (Li et al., 2024c) adopt a failure-driven mechanism: attempt RAG first, and switch to LC only if the model outputs "unanswerable". While simple and sometimes effective, this approach has some drawbacks: it is passive, relying solely on re-

trieval failure signals; it incurs redundant overhead, since RAG retrievals must always be executed; it depends on the model's self-assessment, which may be either over-conservative or overconfident; and it offers little interpretability regarding routing decisions (see Appendix 6.6).

To overcome these issues, we introduce **Pre-Route**, a novel proactive routing framework. Our core insight is that LLMs already possess a latent "routing potential"—the ability to analyze tasks, assess coverage, and anticipate information needs—which is underutilized in conventional pipelines. Pre-Route activates this ability through a structured reasoning process *before* generating an answer. Using low-cost metadata such as document type, title, length, and initial snippets, the model proactively performs: task type analysis, context coverage assessment, information need prediction, and routing choice. Furthermore, this structured reasoning process can be distilled into smaller models, enabling better accuracy with more lightweight deployment. Code is available here.

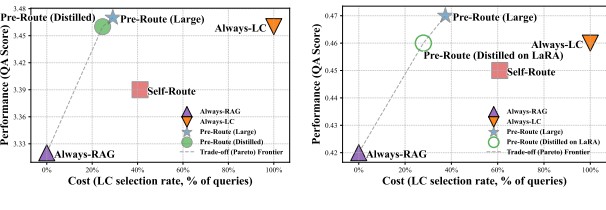

(a) LaRA (in-domain)   (b) LongBench-v2 (OOD)

Figure 1: Visualization of Pre-Route cost-effectiveness with Qwen3-235B-A22B: (a) in-domain (LaRA), (b) out-of-domain (LongBench-v2).

In summary, our contributions are:

1. We propose Pre-Route, an active and cost-efficient routing framework that introduces structured reasoning before retrieval and answering, implementing a "plan-then-execute" paradigm.

2. We systematically verify and uncover the latent routing capabilities of LLMs. Through BoN experiments and linear probe analysis, we demonstrate the existence of this ability and find that guidelines can effectively activate and stabilize it, enabling single-sample performance to approach the multi-sample upper bound and forming a more linearly separable "optimal route dimension" in the representation space.

3. We successfully transfer this routing capability to smaller models, forging a lightweight, plug-and-play module. We find that while smaller models struggle to acquire this ability through prompting alone, they can effectively learn the Pre-Route reasoning patterns via distillation. This paves the way for efficient, low-cost deployment on edge and client-side devices.

4. We achieve significant advantages in both in-distribution and out-of-distribution (OOD) tasks. On the LaRA Benchmark and LongBench-v2 datasets, Pre-Route demonstrates superior effectiveness and generalization by outperforming Self-Route in both performance and efficiency.

## 2 FROM DORMANT TO ACTIVE: UNLOCKING LATENT ROUTING POTENTIAL

Existing methods like Self-Route (Li et al., 2024c) treat routing as a retrieval fallback—RAG is always attempted first—implicitly assuming LLMs lack intrinsic planning. In this work, we challenge this assumption and ask a more fundamental question: **Do LLMs already possess a "latent routing ability"—the internal competence to discern task requirements and select an appropriate path?** To address this question, we adopt two complementary perspectives: (i) **behavioral experiments** that demonstrate both the existence of such latent ability and its instability, and (ii) **representation analysis** that probes the model's internal mechanisms to reveal how this ability can be activated. Taken together, our argument is that routing ability is not absent but rather dormant, and can be reliably awakened by suitable guidelines. All analyses in this section follow the same experimental setup in Sec. 4.1 and are conducted on the LaRA test split.

### 2.1 BEHAVIORAL EVIDENCE: REVEALING LATENT ABILITY VIA BEST-OF-N SAMPLING

To disentangle whether routing failures arise from a lack of capacity or merely insufficient activation, we recast routing as a binary classification task (predicting RAG vs. LC). We adopt the **Best-of-N (BoN) sampling** methodology widely used for testing latent abilities.

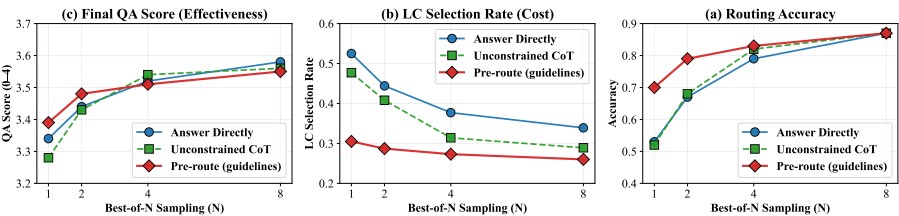

Figure 2: BoN comparison results tested with answer model Qwen3-235B-A22B.

We design three controlled comparisons under identical metadata: (1) Prompt Paradigm: *answer directly*, *unconstrained CoT* (free-form reasoning before routing), and our proposed *Pre-Route* (structured reasoning guidelines). (2) Sampling Strategy: $N \in \{1, 2, 4, 8\}$. (3) Metrics: QA Score, LC selection rate and routing accuracy. We report **routing accuracy**, defined as the proportion of model decisions that match an *ideal label*—the decision rule that maximizes QA performance while defaulting to the lower-cost RAG option when performances are comparable (formalized later in Sec. 3.4).

Experimental results (Fig. 2) clearly reveal the model's internal decision capacity. Under *Answer Directly* and *Unconstrained CoT*, routing accuracy shows a strong positive correlation with $N$. For instance, in the Answer Directly setting, accuracy rises from about 0.53 (N=1) to 0.87 (N=8), indicating that the knowledge required for correct routing is present but accessed stochastically: sometimes the model follows the correct path, other times it diverges. Increasing the number of samples simply raises the chance of hitting the right path, thereby exposing the model's latent ability.

In sharp contrast, introducing the *Pre-Route (guidelines)* structured reasoning chain fundamentally changes the picture. Its curve is much flatter: accuracy already reaches 0.70 at N=1—substantially higher than the other two settings—and further climbs to 0.83 at N=4 before saturating at 0.87 by N=8. This pattern indicates that **structured guidance acts as a calibrator and stabilizer**. It does not inject new knowledge, but rather provides a clear reasoning scaffold that reliably activates and directs the model's latent routing capability, enabling single-shot performance close to the model's upper bound.

## 2.2 REPRESENTATION EVIDENCE: PROBING THE RESHAPED DECISION SPACE

While behavioral results confirm that routing ability can be activated, an open question is what changes occur internally. We hypothesize that structured prompting not only alters outputs, but also reshapes the representation space, such that routing signals become more linearly separable. To investigate this, we apply linear probes on frozen representations and examine four prediction targets: (i) the *ideal* decision, defined in Sec. 3.4 as the

| Model | Prompt | Distilled | Accuracy | | | |
|---|---|---|---|---|---|---|
| | | | Ideal | Route | Doc | Task |
| Qwen-3-1.7B | Pre-Route | ✓ | **0.6389** | **0.7986** | **0.3958** | 0.4097 |
| Qwen-3-1.7B | Pre-Route | ✗ | 0.6250 | 0.7639 | 0.3330 | **0.4167** |
| Qwen-3-8B | answer direct. | ✗ | 0.5486 | 0.6597 | 0.2986 | 0.2569 |
| Qwen-3-1.7B | answer direct. | ✗ | 0.5208 | 0.6389 | 0.3194 | 0.2500 |
| Qwen-3-8B | unconstr. CoT | ✗ | 0.4653 | 0.6944 | 0.3681 | 0.2500 |
| Qwen-3-1.7B | unconstr. CoT | ✗ | 0.3958 | 0.5764 | 0.3472 | 0.2639 |

Table 1: Linear probe accuracy tested with answer model Qwen3-235B-A22B.

*task-optimal decision*—the choice that maximizes QA performance while defaulting to RAG when tied; (ii) the model's own *route* choice, reflecting its internal bias; (iii) *doc_type* (7 classes), to test whether routing simply mirrors document categories; and (iv) *task_type* (4 classes), to test whether routing is merely driven by shallow task-level hints rather than deeper reasoning signals. Among these, ideal label accuracy is the most important indicator, since it directly evaluates whether the model's latent routing aligns with task-optimal decisions.

We apply *linear probing* (Alain & Bengio, 2017): a linear classifier trained on frozen penultimate last token embeddings to predict the labels above. Linear probes are deliberately chosen since linear separability implies explicit encoding in the representation. 1) **Prompt structure is critical:** on Qwen3-1.7B, ideal-label accuracy improves from 0.396 (unconstr. CoT) to 0.521 (direct) to **0.625** (Pre-Route). 2) **Distillation further aligns:** adding distillation(see Sec. 3.4) on Pre-Route raises accuracy to 0.639. 3) **Design over scale:** Qwen3-8B with direct prompting (0.549) underperforms Qwen3-1.7B with Pre-Route (0.625). 4) **Model's own routing decision:** under structured guidance, the model's representation of its own decisions becomes clearer, making probes better at predicting its *route* choice (0.58–0.80), though not always aligned with the task-optimal choice. 5) **Not shal-**

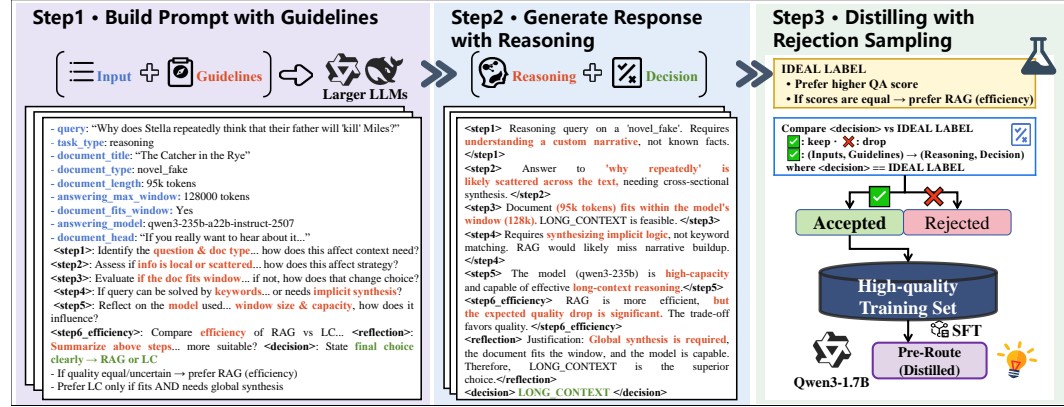

Figure 3: Overview of the Pre-Route framework. Step 1 builds structured prompts with meta-infos and guidelines; Step 2 generates reasoning-based routing decisions; Step 3 performs rejection sampling to filter out suboptimal decisions, distilling high-quality training data for efficient and interpretable routing with smaller models (Qwen3-1.7B). (Complete prompt is provided in Fig. 9)

**low heuristics:** low accuracy on *doc_type* and *task_type* shows routing is not reducible to trivial document or task categories.

Together, these findings establish that LLMs possess a latent but activatable routing ability. BoN experiments demonstrate it behaviorally, while probing confirms it representationally. Crucially, structured guidance and light supervision reliably awaken and stabilize this ability—laying the foundation for the Pre-Route framework introduced next.

# 3 METHODOLOGY

## 3.1 PROBLEM FORMALIZATION AND THEORETICAL FRAMEWORK

**Decision-theoretic formalization.** Given a query $q$ and long document $\mathcal{D}$, a routing policy $\pi$ selects $y \in \{RAG, LC\}$ solely from low-cost meta-information $m$ (title, type, length estimate, leading snippet, etc.), leveraging the observation that $m$ often encodes sufficient cues about information distribution and task difficulty. Formally, we let $U(y; q, \mathcal{D})$ denote the expected QA performance of decision $y$, and $C(y; q, \mathcal{D})$ denote the corresponding execution cost.

## 3.2 THE PRE-ROUTE FRAMEWORK

**Core hypothesis and idea.** LLMs already encode task understanding and information-need sensing, yet standard pipelines do not explicitly *activate* this capacity. Pre-Route triggers the latent planning ability by performing **structured reasoning with decision labels** *before* answering: (i) a **pre-decision** is made prior to costly full-document processing; (ii) the decision relies on **meta-information only**, avoiding full-document retrieval overhead; (iii) a **structured rationale** provides interpretability. In practice, Pre-Route uses only low-cost $m$: user query, task type, document title/type, document length, answering model, a short document head, and RAG configuration; no retriever or answering model is called, markedly reducing planning overhead $C_{\text{route}}$ (see cost analysis below).

**Structured reasoning-chain design.** Routing is conceptually grounded in four broad *decision dimensions*—(i) task & document characterization, i.e., how task or document types affect the need for retrieval vs. long-context; (ii) information distribution, i.e., whether relevant content is missing, fragmented, or dispersed across sections; (iii) capability–constraint matching, i.e., whether retrieval suffices or long-context is required to avoid positional or window-size limits; and (iv) decision with explanation, i.e., outputting the route together with rationale and fallback considerations. In practice, these dimensions are instantiated through six concise *steps*: **(1) task & document characterization**; **(2) distribution pattern judgment**; **(3) context-window feasibility**; **(4) retrieval feasibility**; **(5) model capability consideration**; and **(6) efficiency trade-off** (see Fig. 3). This layered design

grounds the router in theoretical completeness while highlighting operational practicality, thereby yielding interpretability, data efficiency (no large discriminators), and robust generalization.

### 3.3 Cost Analysis

We decompose cost into a routing part and an answering part:

$$C(y; q, \mathcal{D}) = C_{\text{route}}(m) + C_{\text{answer}}(y; q, \mathcal{D}), \tag{1}$$

where $C_{\text{route}}(m)$ is the cost of making route decision, and $C_{\text{answer}}(y; q, \mathcal{D})$ corresponds to downstream execution (retrieval for RAG vs. long-context processing for LC). Using this decomposition, the expected total cost of a routing policy with LC-selection probability $p(\text{LC}) = \Pr(y = LC)$ is

$$C = C_{\text{route}} + p(\text{LC}) \cdot C_{\text{LC}} + \big(1 - p(\text{LC})\big) \cdot C_{\text{RAG}}, \tag{2}$$

where $C_{\text{LC}}$ and $C_{\text{RAG}}$ denote the average answering costs under LC and RAG respectively. In what follows we analyze $C_{\text{route}}$ and $C_{\text{answer}}$ separately and then combine the two perspectives.

#### 3.3.1 Analysis of $C_{\text{route}}$: Self-Route vs. Pre-Route

Pre-Route achieves lower routing cost than the Self-Route baseline once distillation is applied: with Qwen3-1.7B the cost is reduced to about one-fifth that of Self-Route (Tab. 2). With Qwen3-235B, the cost is slightly higher due to the generated rationale, but this overhead is offset once a smaller router is used. This demonstrates that planning overhead can be kept low while retaining interpretability.

Table 2: Per-decision routing cost in USD (average query, computed over all queries in the LaRA benchmark).

| Method | Model | Input Toks | Output Toks | Cost ($\times 10^{-3}$ USD) | | |
| --- | --- | --- | --- | --- | --- | --- |
| | | | | Input | Output | **Total** |
| Self-Route | Qwen3-235B | 2600 | 27 | 0.73 | 0.03 | 0.76 |
| Pre-Route | Qwen3-235B | 1205 | 648 | 0.34 | 0.73 | 1.07 |
| Pre-Route | Qwen3-1.7B | 1205 | 670 | 0.05 | 0.11 | **0.16** |

#### 3.3.2 Analysis of $C_{\text{answer}}$: LC vs. RAG

Answering cost is the dominant component, and it is largely governed by the LC selection probability $p(\text{LC})$. A long-context (LC) pass processes the entire document, whereas retrieval-augmented generation (RAG) operates only on retrieved passages. Since $C_{\text{LC}}$ is substantially larger than $C_{\text{RAG}}$, even moderate changes in $p(\text{LC})$ have a significant impact on total cost. Hence, controlling $p(\text{LC})$ is central to cost management.

#### 3.3.3 Overall cost comparison

When both components are combined, routing overhead $C_{\text{route}}$ remains small relative to answering cost. Even with a large router (235B), planning cost is below 4% of a single 100k-token LC pass ($0.028); with a small router (1.7B), the proportion falls below 1%. These comparisons indicate that the total cost is primarily determined by the answering term, and in particular by the LC selection probability $p(\text{LC})$.

### 3.4 Learning Paradigm: Selective Strategy Alignment via Distillation

**Motivation and cost optimization.** The overall cost is primarily determined by the answering component, and in particular by the LC selection probability $p(\text{LC})$. Yet Routing overhead $C_{\text{route}}$ still matters for latency and scalability in practical deployment. Large models (e.g., Qwen3-235B) can produce strong routing in zero-shot settings, yet the planning overhead $C_{\text{route}}$ is non-trivial. We distill the planning ability into smaller models (e.g., Qwen-7B/1.7B) to reduce cost and latency, improve deployability in constrained environments, and specialize the student for routing.

**Learning targets.** For fixed meta-information $m$, the objective is to maximize expected QA performance by choosing the option with higher performance. When utilities are comparable or the difference is marginal (e.g., tied scores in multi-level or binary evaluation), the lower-cost RAG is preferred, since $C_{\text{RAG}} \ll C_{\text{LC}}$. This principle leads directly to the definition of the **ideal label** for each sample:

$$\hat{y}_{\text{ideal}} = \begin{cases} LC, & U(LC; q, \mathcal{D}) > U(RAG; q, \mathcal{D}), \\ RAG, & \text{otherwise.} \end{cases} \tag{3}$$

Intuitively, LC is chosen only when its performance advantage over RAG; when utilities are comparable, the lower-cost RAG is preferred. Improving routing accuracy reduces expected performance loss by avoiding wasteful LC on unhelpful samples and reallocating budget to cases that truly benefit from LC. We formally prove in Appendix 6.2 that optimizing the ideal label ensures minimal expected cost among all strategies consistent with performance-optimal decisions.

**Stage 1: Rejection sampling.** Let the teacher $\pi_T$ produce candidates $S_i = (T_i, y_i)$; keep entries where the teacher decision matches the ideal rule,

$$\mathcal{D}_{\text{filtered}} = \{(m_i, T_i, y_i) \mid y_i = \hat{y}_{\text{ideal},i}\}, \tag{4}$$

where $m_i$, $T_i$, $y_i$, and $\hat{y}_{\text{ideal},i}$ denote the metadata, reasoning trace, decision, and ideal decision for example $i$, thus ensuring data quality, correcting the empirical distribution toward a near-optimal support, removing teacher errors, and enforcing consistency with task-optimal supervision.

**Stage 2: Path SFT (student fitting on successful support).** Train the student on $\mathcal{D}_{\text{filtered}}$ by minimizing

$$\mathcal{L}_{\text{SFT}}(\theta_S) = -\mathbb{E}_{(m,T,y)\sim\mathcal{D}_{\text{filtered}}}\big[\log \pi_S(T, y \mid m)\big], \tag{5}$$

which is equivalent to minimizing $\text{KL}(q^*\|\pi_S)$ where $q^*$ is the teacher's *successful* sub-distribution. Unlike answer distillation, this transfers *how to reason* rather than merely *what to answer*.

**Data construction strategy.** We generate reasoning chains with Qwen3-235B-A22B and DeepSeek-R1; assign $\hat{y}_{\text{ideal}}$ consistently; enhance scenario diversity by mixing answering-model scales; perform a global 70/10/20 train/val/test split stratified by (type, level, context length); then merge and retain only training entries where the teacher is consistent with the ideal label.

# 4 EXPERIMENTS

## 4.1 EXPERIMENTAL SETUP

**Datasets and Benchmarks** We evaluate Pre-Route on both in-domain and out-of-domain settings. The in-domain dataset is **LaRA**, used for training and primary evaluation. To assess generalization, we further evaluate on the out-of-domain benchmark **LongBench-v2**. For **LaRA**, since knowledge distillation is involved during data construction (see Sec. 3.4), we adopt stratified random splits by *document type*, *task level*, and *context length* for all experiments to ensure fair comparison and prevent leakage. For **LongBench-v2**, we report results on the full evaluation set. Detailed statistics and examples are provided in Appendix 6.3. For retrieval, we follow the default LaRA configuration, using a chunk size of 600 with 100-token overlap, the `gte-multilingual-base` embedding model and `gte-multilingual-reranker-base` reranker (Zhang et al., 2024a), with vector–rerank split ratio 0.5 and rerank size 5.

**Evaluation Metrics** We report three metrics: (i) **Route Accuracy** — the fraction of routing decisions that match the *ideal label* (formalized in Sec. 3.4). The ideal label selects the decision $y \in \{RAG, LC\}$ that maximizes QA performance $U$; in ties, it defaults to $RAG$ to prefer lower cost. Thus, higher Route Accuracy reflects better effectiveness–efficiency alignment. (ii) **QA Score** — downstream answer quality. Because the original LaRA binary metric is too coarse to distinguish superficial vs. substantively complete answers, we adopt a 4-point rubric for finer resolution (examples in Appendix 6.5). (iii) **LC Selection Rate** — the proportion of queries routed to the costly $LC$ path, used as a proxy for computational cost.

**Baselines** We compare against: (a) Always-RAG and Always-LC (fixed strategies); (b) Self-Route (Li et al., 2024c) using the original prompt (Appendix 6.7); and (c) Pre-Route in three variants that we report separately throughout: Pre-Route (Large: DeepSeek-R1, Qwen3-235B, prompt-only), Pre-Route (Small: Qwen3-1.7B, prompt-only), and Pre-Route (Distilled: Distilled-Qwen3-1.7B). All Pre-Route prompts are fixed across models. For all applicable methods we report both thinking **[T]** and no-thinking **[N]** modes. Always-RAG is excluded from LC Rate ranking (LC=0%), while Always-LC is excluded from QA ranking due to prohibitive computational costs. Red-shaded entries denote our recommended **Pre-Route(Large)** and **Pre-Route(Distilled)**.

Table 3: LaRA (In-distribution main experiment): Pre-Route vs. Self-Route and fixed baselines; Best and Second per answer model. *Pre-Route Variants:* Large: R1 (DeepSeek-R1) or Q235B (Qwen3-235B); Small: Q1.7B (Qwen3-1.7B); Distilled: D-Q1.7B (Distilled-Qwen3-1.7B).

| Answer Model
Router Model | Qwen3-1.7B [N] | | | Qwen3-1.7B [T] | | | Qwen3-4B [N] | | | Qwen3-4B [T] | | | Qwen3-8B [N] | | | Qwen3-8B [T] | | |
|---|---|---|---|---|---|---|---|---|---|---|---|---|---|---|---|---|---|---|
| | QA↑ | LC(%)↓ | Acc↑ | QA↑ | LC(%)↓ | Acc↑ | QA↑ | LC(%)↓ | Acc↑ | QA↑ | LC(%)↓ | Acc↑ | QA↑ | LC(%)↓ | Acc↑ | QA↑ | LC(%)↓ | Acc↑ |
| Always-LC (Baseline) | 2.13 | 100.0 | 0.18 | 2.29 | 100.0 | 0.18 | 2.89 | 100.0 | 0.31 | 3.14 | 100.0 | 0.36 | 2.98 | 100.0 | 0.31 | 3.23 | 100.0 | 0.35 |
| Always-RAG (Baseline) | 2.70 | 0.0 | 0.82 | 2.89 | 0.0 | 0.82 | 3.08 | 0.0 | 0.69 | 3.17 | 0.0 | 0.64 | 3.13 | 0.0 | 0.69 | 3.22 | 0.0 | 0.65 |
| Self-Route (Baseline) | 2.22 | 33.6 | 0.49 | 2.40 | 31.7 | 0.48 | 2.81 | 24.6 | 0.53 | 3.06 | 30.5 | 0.47 | 3.04 | 28.1 | 0.55 | 3.23 | 30.7 | 0.57 |
| Pre-Route (R1) [T] | 2.70 | 2.7 | 0.83 | 2.88 | 1.8 | 0.83 | 3.09 | 5.1 | 0.73 | 3.20 | 4.8 | 0.70 | 3.15 | 5.0 | 0.76 | 3.23 | 8.0 | 0.70 |
| Pre-Route (Q235B) [N] | 2.71 | 4.7 | 0.83 | 2.90 | 4.8 | 0.82 | 3.13 | 21.1 | 0.73 | 3.26 | 26.6 | 0.70 | 3.16 | 22.2 | 0.73 | 3.30 | 26.2 | 0.70 |
| Pre-Route (Q235B) [T] | 2.71 | 2.3 | 0.84 | 2.91 | 3.0 | 0.84 | 3.11 | 13.1 | 0.73 | 3.22 | 15.7 | 0.70 | 3.17 | 15.9 | 0.74 | 3.27 | 17.9 | 0.71 |
| Pre-Route (Q1.7B) [N] | 2.70 | 7.9 | 0.79 | 2.85 | 10.8 | 0.77 | 3.04 | 30.1 | 0.61 | 3.20 | 35.9 | 0.57 | 3.12 | 34.0 | 0.60 | 3.29 | 33.2 | 0.63 |
| Pre-Route (Q1.7B) [T] | 2.68 | 10.8 | 0.77 | 2.87 | 11.1 | 0.80 | 3.07 | 30.8 | 0.62 | 3.22 | 40.1 | 0.58 | 3.10 | 35.6 | 0.61 | 3.25 | 39.9 | 0.58 |
| Pre-Route (D-Q1.7B) [N] | 2.70 | 3.6 | 0.82 | 2.89 | 3.9 | 0.83 | 3.10 | 17.9 | 0.71 | 3.26 | 21.2 | 0.71 | 3.16 | 21.5 | 0.73 | 3.30 | 21.4 | 0.71 |
| Pre-Route (D-Q1.7B) [T] | 2.71 | 3.2 | 0.83 | 2.89 | 2.8 | 0.82 | 3.10 | 21.1 | 0.72 | 3.26 | 20.5 | 0.70 | 3.19 | 19.5 | 0.76 | 3.28 | 20.8 | 0.70 |

| Answer Model
Router Model | Qwen3-30B [N] | | | Qwen3-30B [T] | | | Qwen3-235B [N] | | | Qwen3-235B [T] | | | DeepSeek-R1 [T] | | | Qwen-Max [N] | | |
|---|---|---|---|---|---|---|---|---|---|---|---|---|---|---|---|---|---|---|
| | QA↑ | LC(%)↓ | Acc↑ | QA↑ | LC(%)↓ | Acc↑ | QA↑ | LC(%)↓ | Acc↑ | QA↑ | LC(%)↓ | Acc↑ | QA↑ | LC(%)↓ | Acc↑ | QA↑ | LC(%)↓ | Acc↑ |
| Always-LC (Baseline) | 3.37 | 100.0 | 0.40 | 3.39 | 100.0 | 0.39 | 3.46 | 100.0 | 0.34 | 3.51 | 100.0 | 0.40 | 3.44 | 100.0 | 0.35 | 3.36 | 100.0 | 0.39 |
| Always-RAG (Baseline) | 3.18 | 0.0 | 0.60 | 3.27 | 0.0 | 0.61 | 3.32 | 0.0 | 0.66 | 3.33 | 0.0 | 0.60 | 3.38 | 0.0 | 0.65 | 3.20 | 0.0 | 0.61 |
| Self-Route (Baseline) | 3.20 | 33.9 | 0.53 | 3.10 | 35.5 | 0.42 | 3.39 | 41.1 | 0.56 | 3.34 | 33.9 | 0.52 | 3.36 | 31.4 | 0.52 | 3.28 | 36.5 | 0.56 |
| Pre-Route (R1) [T] | 3.26 | 14.9 | 0.69 | 3.30 | 9.6 | 0.69 | 3.39 | 14.2 | 0.73 | 3.37 | 10.8 | 0.68 | 3.42 | 10.0 | 0.70 | 3.24 | 10.6 | 0.67 |
| Pre-Route (Q235B) [N] | 3.31 | 33.8 | 0.65 | 3.39 | 29.4 | 0.68 | 3.47 | 29.1 | 0.72 | 3.43 | 27.2 | 0.67 | 3.47 | 27.2 | 0.70 | 3.31 | 23.1 | 0.69 |
| Pre-Route (Q235B) [T] | 3.27 | 17.9 | 0.68 | 3.32 | 18.6 | 0.69 | 3.42 | 20.2 | 0.72 | 3.40 | 18.3 | 0.68 | 3.45 | 16.9 | 0.71 | 3.29 | 19.6 | 0.69 |
| Pre-Route (Q1.7B) [N] | 3.26 | 42.3 | 0.53 | 3.37 | 37.2 | 0.57 | 3.36 | 40.9 | 0.53 | 3.41 | 40.6 | 0.54 | 3.44 | 32.5 | 0.56 | 3.25 | 36.6 | 0.55 |
| Pre-Route (Q1.7B) [T] | 3.27 | 40.0 | 0.61 | 3.36 | 38.5 | 0.59 | 3.41 | 35.6 | 0.59 | 3.41 | 33.1 | 0.59 | 3.49 | 31.5 | 0.68 | 3.26 | 28.3 | 0.60 |
| Pre-Route (D-Q1.7B) [N] | 3.28 | 27.7 | 0.66 | 3.39 | 28.0 | 0.70 | 3.46 | 24.6 | 0.74 | 3.43 | 22.7 | 0.69 | 3.51 | 20.6 | 0.73 | 3.28 | 24.7 | 0.67 |
| Pre-Route (D-Q1.7B) [T] | 3.29 | 26.4 | 0.65 | 3.35 | 24.3 | 0.67 | 3.47 | 26.9 | 0.73 | 3.44 | 26.0 | 0.67 | 3.47 | 20.3 | 0.72 | 3.31 | 24.5 | 0.69 |

**Models** Our experiments employ the latest available models from the recent Qwen3 series, spanning the 235B, 30B, 8B, 4B, and 1.7B scales. Qwen3-235B and Qwen3-30B are from the July 2025 version. Our fine-tuned student model is referred to as Distilled-Qwen3-1.7B. We also include DeepSeek-R1 (May 2025 version) and the proprietary model Qwen-Max (January 2025 version).

## 4.2 Main Results on LaRA Benchmark

Tab. 3 summarizes the LaRA in-domain results. Note that **top-two results (bold/underline) consistently fall in our red shaded Pre-Route methods.** Overall, Pre-Route consistently surpasses Self-Route in QA score and route accuracy across all settings, while also reducing the reliance on LC. proving its robustness across various answer model sizes and thinking modes. We also provide more case studies on why sometimes Self-Route fails on LaRA in Appendix 6.6.

**Large model prompt-only routing.** Using Qwen3-235B and DeepSeek-R1, we find that Pre-Route (Prompt-only) significantly outperforms Self-Route. For instance, with Qwen3-4B [T] as the answer model, DeepSeek-R1 [T] attains much higher QA and accuracy at a fraction of the LC cost; with Qwen3-235B [T] as the answer model, the Qwen3-235B router likewise yields clear gains over Self-Route. Structured reasoning prompts alone can reliably activate latent routing ability, yielding both higher route accuracy and QA score. This demonstrates that well-designed prompts suffice to unlock models' metacognitive planning capabilities.

**Transferability to small models.** When directly applied to Qwen3-1.7B with identical prompts, performance degrades: reasoning chains and routing decisions become unstable. For example, under the Qwen3-235B [T] answer model, Qwen3-1.7B routing lags far behind the large-model routers in accuracy despite comparable QA scores, highlighting that small routers cannot directly inherit complex planning behavior. This indicates that small models cannot directly inherit complex planning behavior and require explicit distillation to reach reliable routing performance.

**Distillation on in-domain data.** Using Pre-Route (Large) as teacher, we distill into a lightweight Qwen3-1.7B student with reasoning-chain supervision on LaRA. The distilled model successfully acquires structured routing logic, approaching the teacher's QA performance while achieving a lower LC selection rate. For instance, with the strongest answer model Qwen-Max [N], the distilled router reaches clearly higher accuracy at substantially reduced LC usage compared to Self-Route. These results verify that large-model "planning intelligence" can be effectively compressed and transferred. Notably, the non-thinking distilled router already performs well, making it the most resource-efficient configuration.

## 4.3 Generalization and Robustness

Tables 4, 5, and 6 report LongBench-v2 under rerank sizes 5/7/10, and Fig. 4 further visualizes how the LC vs. RAG balance shifts as rerank size increases, underlining the importance of testing robustness under different retriever configurations.

**Out-of-distribution evaluation.** Compared to LaRA, LongBench-v2 is out-of-domain not only in data distribution but also in task format and evaluation protocol: LaRA uses open-ended QA with graded scores (0–4), whereas LongBench-v2 reformulates tasks as four-choice multiple-choice questions(MCQ) with binary correctness. This makes Self-Route appear more competitive in QA scores, since its weakness of sometimes routing partially correct answers (as in Appendix 6.6) is no longer penalized—under binary MCQ evaluation the router only needs to ensure correctness rather than differentiate answer quality. Nevertheless, Pre-Route remains more cost-efficient than Self-Route with substantially low LC usage.

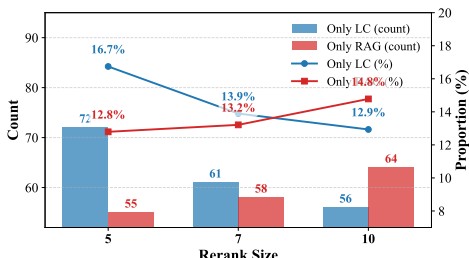

Figure 4: Qwen3-235B: Number of questions answered correctly *only* by LC vs. *only* by RAG.

**Robustness to retrieval configuration.** Across rerank sizes (5/7/10), Pre-Route (Distilled) remains remarkably stable, showing minimal sensitivity to retriever changes. For example, with a *small* answer model such as Qwen3-1.7B, or a *large* one such as Qwen-Max, the distilled router consistently achieves higher accuracy while cutting LC usage by a large margin. Together with the out-of-distribution results, these findings demonstrate that Pre-Route retains its advantage across both task formats and retriever settings, highlighting its robustness for practical deployment.

Table 4: LongBench-v2 (OOD, rerank size = 5): Pre-Route results; **Best** and Second per answer model. *Pre-Route Variants:* Large: R1 (DeepSeek-R1) or Q235B (Qwen3-235B); Small: Q1.7B (Qwen3-1.7B); Distilled: D-Q1.7B (Distilled-Qwen3-1.7B).

| Answer Model | Qwen3-1.7B [N] | | | Qwen3-1.7B [T] | | | Qwen3-4B [N] | | | Qwen3-4B [T] | | | Qwen3-8B [N] | | | Qwen3-8B [T] | | |
| Router Model | QA↑ | LC(%)↓ | Acc↑ | QA↑ | LC(%)↓ | Acc↑ | QA↑ | LC(%)↓ | Acc↑ | QA↑ | LC(%)↓ | Acc↑ | QA↑ | LC(%)↓ | Acc↑ | QA↑ | LC(%)↓ | Acc↑ |
|---|---|---|---|---|---|---|---|---|---|---|---|---|---|---|---|---|---|---|
| Always-LC (Baseline) | 0.29 | 100.0 | 0.38 | 0.28 | 100.0 | 0.31 | 0.30 | 100.0 | 0.30 | 0.37 | 100.0 | 0.40 | 0.40 | 100.0 | 0.40 | 0.40 | 100.0 | 0.37 |
| Always-RAG (Baseline) | 0.33 | 0.0 | 0.62 | 0.34 | 0.0 | 0.69 | 0.31 | 0.0 | 0.70 | 0.34 | 0.0 | 0.60 | 0.34 | 0.0 | 0.60 | 0.37 | 0.0 | 0.63 |
| Self-Route (Baseline) | **0.32** | 34.9 | 0.63 | **0.33** | 28.1 | 0.68 | 0.30 | 22.7 | 0.69 | **0.37** | 33.8 | 0.65 | **0.38** | 31.3 | 0.64 | **0.41** | 32.3 | 0.67 |
| Pre-Route (R1) [T] | 0.30 | 24.8 | 0.68 | **0.33** | 23.5 | 0.71 | 0.31 | 24.7 | 0.69 | 0.35 | 24.5 | 0.70 | 0.37 | 19.6 | **0.72** | 0.39 | **19.9** | **0.71** |
| Pre-Route (Q235B) [N] | 0.31 | 39.7 | 0.58 | 0.31 | 37.8 | 0.60 | 0.31 | 39.5 | 0.59 | 0.36 | 38.7 | 0.59 | **0.38** | 31.8 | 0.64 | 0.40 | 33.0 | 0.63 |
| Pre-Route (Q235B) [T] | 0.31 | 29.7 | 0.66 | 0.32 | 29.4 | 0.65 | **0.32** | 27.0 | 0.69 | 0.34 | 30.1 | 0.64 | 0.37 | 25.8 | 0.68 | 0.38 | 29.6 | 0.63 |
| Pre-Route (Q1.7B) [N] | 0.30 | 37.6 | 0.56 | 0.31 | 38.9 | 0.56 | 0.30 | 41.2 | 0.56 | 0.34 | 38.2 | 0.58 | 0.35 | 39.8 | 0.55 | 0.38 | 38.1 | 0.58 |
| Pre-Route (Q1.7B) [T] | 0.30 | 41.7 | 0.54 | 0.32 | 38.7 | 0.58 | **0.32** | 42.1 | 0.58 | 0.34 | 40.7 | 0.55 | **0.38** | 38.9 | 0.61 | 0.37 | 40.0 | 0.54 |
| Pre-Route (D-Q1.7B) [N] | 0.31 | 7.3 | 0.81 | **0.33** | 8.0 | 0.80 | 0.31 | 20.8 | 0.70 | 0.34 | 21.1 | 0.67 | **0.38** | 20.7 | 0.70 | 0.38 | 21.9 | 0.68 |
| Pre-Route (D-Q1.7B) [T] | **0.32** | **6.6** | **0.82** | **0.33** | **7.5** | 0.80 | 0.31 | 20.2 | **0.71** | 0.34 | 20.4 | 0.66 | 0.36 | 18.9 | 0.69 | 0.36 | 21.2 | 0.68 |

| Answer Model | Qwen3-30B [N] | | | Qwen3-30B [T] | | | Qwen3-235B [N] | | | Qwen3-235B [T] | | | DeepSeek-R1 [T] | | | Qwen-Max [N] | | |
| Router Model | QA↑ | LC(%)↓ | Acc↑ | QA↑ | LC(%)↓ | Acc↑ | QA↑ | LC(%)↓ | Acc↑ | QA↑ | LC(%)↓ | Acc↑ | QA↑ | LC(%)↓ | Acc↑ | QA↑ | LC(%)↓ | Acc↑ |
|---|---|---|---|---|---|---|---|---|---|---|---|---|---|---|---|---|---|---|
| Always-LC (Baseline) | 0.40 | 100.0 | 0.40 | 0.36 | 100.0 | 0.56 | 0.46 | 100.0 | 0.60 | 0.52 | 100.0 | 0.54 | 0.53 | 100.0 | 0.49 | 0.48 | 100.0 | 0.59 |
| Always-RAG (Baseline) | 0.39 | 0.0 | 0.60 | 0.22 | 0.0 | 0.44 | 0.42 | 0.0 | 0.40 | 0.43 | 0.0 | 0.46 | 0.45 | 0.0 | 0.51 | 0.40 | 0.0 | 0.41 |
| Self-Route (Baseline) | 0.40 | 38.0 | 0.61 | 0.28 | 47.8 | 0.51 | 0.45 | 60.9 | 0.44 | **0.49** | 52.2 | 0.52 | 0.46 | 44.4 | 0.53 | **0.45** | 57.4 | 0.46 |
| Pre-Route (R1) [T] | 0.41 | 25.9 | **0.70** | 0.25 | **25.2** | 0.62 | **0.47** | 23.5 | **0.72** | 0.45 | 23.8 | 0.63 | 0.47 | 27.6 | 0.61 | 0.43 | **25.8** | **0.68** |
| Pre-Route (Q235B) [N] | **0.42** | 39.0 | 0.60 | 0.27 | 40.0 | 0.53 | **0.47** | 37.4 | 0.62 | 0.47 | 41.3 | 0.55 | **0.51** | 41.8 | 0.56 | 0.44 | 40.0 | 0.59 |
| Pre-Route (Q235B) [T] | 0.41 | 29.2 | 0.67 | 0.24 | 28.5 | 0.58 | 0.45 | 27.4 | 0.67 | **0.47** | 28.5 | 0.61 | **0.51** | 31.1 | 0.65 | 0.44 | 29.0 | 0.65 |
| Pre-Route (Q1.7B) [N] | 0.38 | 39.9 | 0.57 | 0.27 | 35.3 | 0.58 | 0.45 | 36.7 | 0.61 | 0.47 | 37.1 | 0.55 | 0.47 | 42.3 | 0.53 | **0.45** | 38.7 | 0.60 |
| Pre-Route (Q1.7B) [T] | 0.40 | 41.0 | 0.57 | 0.27 | 41.2 | 0.54 | 0.44 | 40.5 | 0.56 | 0.46 | 43.0 | 0.55 | 0.47 | 41.3 | 0.57 | 0.44 | 41.8 | 0.56 |
| Pre-Route (D-Q1.7B) [N] | **0.42** | 27.6 | 0.68 | 0.27 | 31.2 | 0.61 | 0.46 | 27.9 | 0.67 | 0.47 | 28.7 | 0.61 | 0.46 | **18.4** | 0.64 | 0.44 | 27.3 | 0.66 |
| Pre-Route (D-Q1.7B) [T] | **0.42** | 28.0 | 0.67 | **0.28** | 29.5 | **0.62** | 0.45 | 26.7 | 0.66 | 0.46 | 29.0 | 0.58 | 0.48 | 21.9 | **0.66** | **0.45** | 27.7 | 0.66 |

Table 5: LongBench-v2 (OOD, rerank size = 7): Pre-Route results; **Best** and Second per answer model. *Pre-Route Variants:* Large: R1 (DeepSeek-R1) or Q235B (Qwen3-235B); Small: Q1.7B (Qwen3-1.7B); Distilled: D-Q1.7B (Distilled-Qwen3-1.7B).

| Answer Model | Qwen3-1.7B [N] | | | Qwen3-1.7B [T] | | | Qwen3-4B [N] | | | Qwen3-4B [T] | | | Qwen3-8B [N] | | | Qwen3-8B [T] | | |
| Router Model | QA↑ | LC(%)↓ | Acc↑ | QA↑ | LC(%)↓ | Acc↑ | QA↑ | LC(%)↓ | Acc↑ | QA↑ | LC(%)↓ | Acc↑ | QA↑ | LC(%)↓ | Acc↑ | QA↑ | LC(%)↓ | Acc↑ |
|---|---|---|---|---|---|---|---|---|---|---|---|---|---|---|---|---|---|---|
| Always-LC (Baseline) | 0.29 | 100.0 | 0.33 | 0.28 | 100.0 | 0.31 | 0.30 | 100.0 | 0.28 | 0.36 | 100.0 | 0.34 | 0.40 | 100.0 | 0.33 | 0.40 | 100.0 | 0.34 |
| Always-RAG (Baseline) | 0.34 | 0.0 | 0.67 | 0.34 | 0.0 | 0.69 | 0.33 | 0.0 | 0.72 | 0.38 | 0.0 | 0.66 | 0.39 | 0.0 | 0.67 | 0.39 | 0.0 | 0.66 |
| Self-Route (Baseline) | **0.34** | 31.1 | 0.68 | 0.33 | 26.0 | 0.69 | 0.32 | 22.0 | **0.72** | 0.37 | 30.6 | 0.64 | 0.37 | 24.8 | 0.68 | 0.39 | 28.6 | 0.65 |
| Pre-Route (R1) [T] | 0.32 | 24.8 | 0.71 | 0.33 | 22.5 | 0.72 | **0.34** | 24.4 | 0.69 | **0.41** | 25.9 | 0.69 | 0.38 | 19.5 | 0.71 | **0.42** | 22.7 | **0.71** |
| Pre-Route (Q235B) [N] | 0.32 | 38.1 | 0.61 | **0.34** | 39.1 | 0.61 | 0.33 | 39.4 | 0.57 | 0.39 | 39.6 | 0.59 | 0.38 | 32.2 | 0.63 | 0.41 | 31.3 | 0.64 |
| Pre-Route (Q235B) [T] | 0.32 | 29.7 | 0.68 | 0.33 | 29.4 | 0.69 | **0.34** | 27.6 | 0.67 | 0.39 | 31.0 | 0.65 | 0.38 | 26.2 | 0.66 | 0.41 | 24.0 | 0.70 |
| Pre-Route (Q1.7B) [N] | **0.34** | 37.7 | 0.62 | 0.30 | 36.4 | 0.60 | 0.32 | 39.4 | 0.58 | 0.36 | 36.8 | 0.57 | 0.38 | 40.1 | 0.57 | 0.39 | 41.3 | 0.57 |
| Pre-Route (Q1.7B) [T] | 0.33 | 42.2 | 0.56 | 0.33 | 43.9 | 0.56 | 0.33 | 40.5 | 0.57 | 0.37 | 40.0 | 0.57 | **0.39** | 45.9 | 0.55 | 0.41 | 43.1 | 0.55 |
| Pre-Route (D-Q1.7B) [N] | **0.34** | **6.8** | **0.84** | 0.33 | 7.7 | **0.81** | **0.34** | 23.9 | 0.69 | 0.37 | 22.9 | 0.68 | 0.37 | 19.1 | **0.72** | 0.40 | **20.4** | 0.70 |
| Pre-Route (D-Q1.7B) [T] | 0.33 | 7.0 | 0.83 | 0.33 | 7.7 | **0.81** | 0.33 | 20.3 | 0.69 | 0.38 | 19.7 | **0.71** | 0.36 | 20.4 | 0.69 | 0.40 | 21.5 | 0.69 |

| Answer Model | Qwen3-30B [N] | | | Qwen3-30B [T] | | | Qwen3-235B [N] | | | Qwen3-235B [T] | | | DeepSeek-R1 [T] | | | Qwen-Max [N] | | |
| Router Model | QA↑ | LC(%)↓ | Acc↑ | QA↑ | LC(%)↓ | Acc↑ | QA↑ | LC(%)↓ | Acc↑ | QA↑ | LC(%)↓ | Acc↑ | QA↑ | LC(%)↓ | Acc↑ | QA↑ | LC(%)↓ | Acc↑ |
|---|---|---|---|---|---|---|---|---|---|---|---|---|---|---|---|---|---|---|
| Always-LC (Baseline) | 0.40 | 100.0 | 0.34 | 0.36 | 100.0 | 0.51 | 0.47 | 100.0 | 0.54 | 0.52 | 100.0 | 0.52 | 0.53 | 100.0 | 0.43 | 0.48 | 100.0 | 0.52 |
| Always-RAG (Baseline) | 0.42 | 0.0 | 0.66 | 0.24 | 0.0 | 0.44 | 0.46 | 0.0 | 0.46 | 0.45 | 0.0 | 0.48 | 0.49 | 0.0 | 0.57 | 0.44 | 0.0 | 0.48 |
| Self-Route (Baseline) | 0.42 | 29.5 | 0.66 | 0.28 | 40.6 | 0.54 | 0.46 | 57.2 | 0.46 | **0.50** | 46.6 | 0.55 | 0.54 | 35.4 | 0.62 | 0.46 | 49.7 | 0.51 |
| Pre-Route (R1) [T] | 0.40 | **25.2** | **0.67** | 0.25 | **25.5** | 0.62 | **0.49** | 25.3 | **0.72** | 0.48 | 24.0 | 0.65 | 0.53 | 26.2 | 0.70 | 0.46 | 25.8 | **0.69** |
| Pre-Route (Q235B) [N] | 0.40 | 36.8 | 0.58 | 0.28 | 35.1 | 0.60 | **0.49** | 39.0 | 0.62 | **0.50** | 38.1 | 0.59 | **0.55** | 40.3 | 0.62 | **0.47** | 39.1 | 0.61 |
| Pre-Route (Q235B) [T] | 0.41 | 29.2 | 0.66 | 0.26 | 28.8 | 0.60 | **0.49** | 29.2 | 0.68 | 0.48 | 28.3 | 0.63 | 0.54 | 36.4 | 0.62 | **0.47** | 29.8 | 0.67 |
| Pre-Route (Q1.7B) [N] | 0.40 | 48.1 | 0.51 | 0.29 | 42.2 | 0.53 | 0.47 | 36.0 | 0.61 | 0.47 | 35.7 | 0.57 | 0.53 | 41.3 | 0.56 | 0.46 | 42.1 | 0.56 |
| Pre-Route (Q1.7B) [T] | 0.40 | 44.8 | 0.52 | **0.30** | 43.9 | 0.56 | 0.46 | 34.4 | 0.60 | **0.50** | 40.5 | 0.57 | 0.53 | 40.8 | 0.59 | 0.45 | 42.3 | 0.54 |
| Pre-Route (D-Q1.7B) [N] | **0.43** | 26.9 | **0.67** | 0.27 | 28.5 | 0.62 | 0.47 | 26.4 | 0.67 | **0.50** | 28.8 | 0.61 | 0.52 | 18.9 | 0.67 | 0.46 | 25.2 | **0.69** |
| Pre-Route (D-Q1.7B) [T] | 0.42 | 28.3 | 0.65 | 0.28 | 27.4 | **0.64** | 0.46 | 28.0 | 0.66 | 0.48 | 26.9 | 0.62 | 0.52 | **17.5** | 0.69 | 0.44 | **24.4** | 0.68 |

Table 6: LongBench-v2 (OOD, rerank size = 10): Pre-Route results; **Best** and Second per answer model. *Pre-Route Variants:* Large: R1 (DeepSeek-R1) or Q235B (Qwen3-235B); Small: Q1.7B (Qwen3-1.7B); Distilled: D-Q1.7B (Distilled-Qwen3-1.7B).

| Answer Model / Router Model | Qwen3-1.7B [N] | | | Qwen3-1.7B [T] | | | Qwen3-4B [N] | | | Qwen3-4B [T] | | | Qwen3-8B [N] | | | Qwen3-8B [T] | | |
|---|---|---|---|---|---|---|---|---|---|---|---|---|---|---|---|---|---|---|
| | QA↑ | LC(%)↓ | Acc↑ | QA↑ | LC(%)↓ | Acc↑ | QA↑ | LC(%)↓ | Acc↑ | QA↑ | LC(%)↓ | Acc↑ | QA↑ | LC(%)↓ | Acc↑ | QA↑ | LC(%)↓ | Acc↑ |
| Always-LC (Baseline) | 0.29 | 100.0 | 0.32 | 0.27 | 100.0 | 0.30 | 0.30 | 100.0 | 0.26 | 0.36 | 100.0 | 0.30 | 0.40 | 100.0 | 0.27 | 0.40 | 100.0 | 0.28 |
| Always-RAG (Baseline) | 0.32 | 0.0 | 0.68 | 0.31 | 0.0 | 0.70 | 0.33 | 0.0 | 0.74 | 0.39 | 0.0 | 0.70 | 0.43 | 0.0 | 0.73 | 0.43 | 0.0 | 0.72 |
| Self-Route (Baseline) | **0.33** | 28.2 | 0.69 | 0.31 | 24.2 | 0.70 | 0.33 | 19.5 | 0.74 | 0.38 | 26.0 | **0.68** | 0.41 | 21.1 | 0.72 | 0.41 | 24.3 | 0.70 |
| Pre-Route (R1) [T] | 0.31 | 25.9 | 0.68 | **0.32** | 24.8 | 0.71 | 0.33 | 23.6 | 0.70 | **0.40** | 29.5 | 0.64 | 0.44 | **19.3** | **0.74** | 0.45 | 20.9 | **0.76** |
| Pre-Route (Q235B) [N] | 0.31 | 39.5 | 0.58 | 0.31 | 40.2 | 0.59 | 0.33 | 38.6 | 0.57 | 0.39 | 40.1 | 0.57 | **0.45** | 31.7 | 0.66 | **0.46** | 32.8 | 0.67 |
| Pre-Route (Q235B) [T] | 0.32 | 29.7 | 0.66 | 0.31 | 27.5 | 0.68 | **0.34** | 29.4 | 0.66 | 0.39 | 30.4 | 0.64 | 0.43 | 27.4 | 0.66 | 0.45 | 27.5 | 0.70 |
| Pre-Route (Q1.7B) [N] | **0.33** | 39.9 | 0.59 | 0.31 | 34.1 | 0.62 | 0.32 | 35.8 | 0.58 | 0.36 | 38.0 | 0.56 | 0.41 | 40.0 | 0.56 | 0.42 | 36.5 | 0.61 |
| Pre-Route (Q1.7B) [T] | 0.31 | 40.3 | 0.56 | 0.31 | 41.1 | 0.57 | 0.33 | 41.6 | 0.56 | 0.38 | 38.9 | 0.56 | 0.43 | 39.1 | 0.60 | 0.43 | 42.2 | 0.57 |
| Pre-Route (D-Q1.7B) [N] | **0.33** | 7.6 | **0.83** | 0.31 | 7.5 | **0.82** | **0.34** | 21.2 | 0.71 | 0.38 | 23.7 | **0.68** | 0.42 | 23.4 | 0.71 | 0.42 | 21.6 | 0.69 |
| Pre-Route (D-Q1.7B) [T] | 0.32 | **6.8** | **0.83** | 0.31 | 9.5 | 0.80 | 0.32 | 20.6 | 0.70 | 0.37 | 22.4 | 0.66 | 0.42 | 22.0 | 0.72 | 0.44 | **20.9** | 0.71 |

| Answer Model / Router Model | Qwen3-30B [N] | | | Qwen3-30B [T] | | | Qwen3-235B [N] | | | Qwen3-235B [T] | | | DeepSeek-R1 [T] | | | Qwen-Max [N] | | |
|---|---|---|---|---|---|---|---|---|---|---|---|---|---|---|---|---|---|---|
| | QA↑ | LC(%)↓ | Acc↑ | QA↑ | LC(%)↓ | Acc↑ | QA↑ | LC(%)↓ | Acc↑ | QA↑ | LC(%)↓ | Acc↑ | QA↑ | LC(%)↓ | Acc↑ | QA↑ | LC(%)↓ | Acc↑ |
| Always-LC (Baseline) | 0.41 | 100.0 | 0.31 | 0.36 | 100.0 | 0.46 | 0.46 | 100.0 | 0.50 | 0.51 | 100.0 | 0.49 | 0.53 | 100.0 | 0.38 | 0.48 | 100.0 | 0.47 |
| Always-RAG (Baseline) | 0.45 | 0.0 | 0.69 | 0.25 | 0.0 | 0.54 | 0.48 | 0.0 | 0.50 | 0.45 | 0.0 | 0.51 | 0.49 | 0.0 | 0.62 | 0.48 | 0.0 | 0.53 |
| Self-Route (Baseline) | 0.46 | 29.0 | 0.70 | **0.30** | 35.0 | 0.60 | 0.48 | 54.0 | 0.49 | **0.50** | 44.0 | 0.57 | 0.50 | 28.9 | 0.63 | 0.48 | 46.2 | 0.53 |
| Pre-Route (R1) [T] | 0.45 | **22.8** | **0.73** | 0.27 | **25.0** | **0.64** | **0.51** | 25.6 | **0.73** | 0.47 | **23.3** | **0.66** | 0.51 | 24.5 | **0.69** | 0.49 | 25.8 | **0.70** |
| Pre-Route (Q235B) [N] | 0.46 | 36.3 | 0.62 | 0.29 | 37.9 | 0.58 | **0.51** | 38.3 | 0.63 | 0.48 | 40.2 | 0.57 | 0.51 | 39.7 | 0.58 | **0.50** | 39.6 | 0.60 |
| Pre-Route (Q235B) [T] | 0.45 | 29.7 | 0.67 | 0.26 | 31.1 | 0.58 | **0.51** | 26.3 | 0.71 | 0.48 | 31.0 | 0.61 | 0.50 | 32.4 | 0.61 | **0.50** | 30.8 | 0.67 |
| Pre-Route (Q1.7B) [N] | 0.45 | 48.0 | 0.53 | **0.30** | 42.3 | 0.57 | 0.50 | 35.8 | 0.62 | **0.50** | 36.7 | 0.60 | **0.53** | 41.7 | 0.59 | 0.48 | 38.7 | 0.59 |
| Pre-Route (Q1.7B) [T] | 0.46 | 43.2 | 0.57 | **0.30** | 41.1 | 0.57 | 0.50 | 42.3 | 0.57 | 0.49 | 39.5 | 0.56 | **0.53** | 43.6 | 0.52 | 0.48 | 41.1 | 0.57 |
| Pre-Route (D-Q1.7B) [N] | 0.46 | 25.9 | 0.71 | 0.29 | 27.6 | 0.63 | 0.50 | 27.5 | 0.70 | **0.50** | 30.0 | 0.62 | **0.53** | 23.0 | 0.66 | 0.48 | **23.9** | 0.69 |
| Pre-Route (D-Q1.7B) [T] | **0.47** | 25.2 | 0.72 | 0.28 | 26.9 | 0.61 | **0.51** | 27.5 | 0.70 | 0.48 | 29.0 | 0.61 | 0.52 | **19.1** | **0.69** | 0.48 | 25.6 | 0.69 |

## 4.4 PRE-ROUTE GUIDELINES ABLATION

To understand why the structured reasoning chain is effective, we conduct ablations by removing individual steps. As shown in Table 7, the **full guideline** achieves the best balance: highest accuracy with a moderate LC rate. Removing components leads either to QA Score drops or inflated LC usage, both stemming from incorrect route choices.

**Decision rules:** dropping them keeps QA score similar but causes the router to over-rely on LC, greatly increasing cost and reducing accuracy. **Reflection:** removing it slightly lowers accuracy, suggesting it mainly helps refine borderline cases.

Table 7: Ablation analysis. Removing reasoning steps reduces accuracy or increases LC rate, highlighting the necessity of the full Pre-Route design.

| Variant | QA Score | LC Rate (%) | Acc |
|---|---|---|---|
| **Pre-Route (full)** | **3.38** | 20.7 | **0.68** |
| - No Reflection | 3.33 | 20.8 | 0.65 |
| - No Decision Rules | **3.38** | 45.3 | 0.57 |
| - No Step1 | 3.33 | **10.1** | **0.68** |
| - No Step2 | 3.37 | 27.0 | 0.66 |
| - No Step3 | 3.35 | 18.2 | 0.66 |
| - No Step4 | 3.31 | 17.2 | 0.66 |
| - No Step5 | 3.31 | 21.2 | 0.66 |
| - No Step6 | 3.35 | 24.3 | 0.67 |

**Step 1/2 (task & document characterization / distribution judgment):** without them, the router misjudges when LC is needed, either cutting LC too much or using it inappropriately, breaking the accuracy–efficiency balance. **Step 3–6 (window feasibility, retrieval feasibility, model capability, efficiency trade-off):** each prevents misallocation; omitting them makes the router use LC when RAG suffices (wasting cost) or stick to RAG when LC is required (hurting accuracy). Overall, the six steps work as an integrated control loop. Missing any of them causes systematic routing errors—either wasted cost or lost QA Score—whereas the **full guideline** consistently achieves both.

## 4.5 ROBUSTNESS TO METADATA

Pre-Route is not built on the assumption of a fully curated metadata schema. Instead, it deliberately relies on *low-cost structural signals* that are almost always present in realistic long-context systems. In typical RAG or long-context pipelines, document length and a leading snippet (`doc_head`) are naturally available, whereas coarser attributes such as titles, document types, or task tags often appear only as system by-products (e.g., file names, URLs, folder or repository structure, web page titles, knowledge-base tags) and are exactly the fields that tend to be missing or noisy in practice. We therefore treat length and head as a minimal interface, and focus our analysis on *higher-level metadata* that may or may not be present. Importantly, Pre-Route is not solely based on leading snippet: decisions arise from joint reasoning over the query, document length/structure, and hypothesized information distribution, with the head snippet acting as a *soft prior* rather than a hard rule. Appendix 6.6 also provides a case study illustrating how query cues and distributional reasoning lead Pre-Route to prefer LC, yet the head snippet looks like a local factual snippet of a story.

To assess robustness under missing or imprecise metadata, we keep length and head fixed and vary only the availability of higher-level fields on LaRA, comparing three settings:

- **Full-Meta**: all available fields (length, head, title, document type, task type, answering model).
- **Head-only**: only head and length; higher-level fields such as title and document type are removed.
- **Generated-Meta**: a small auxiliary model (Qwen3-1.7B) takes query and leading snippet as input and generates pseudo title, document type, and task type for Pre-Route.

Table 8 reports QA, LC rate, and routing accuracy for several answer–router pairs under these settings, together with the Self-Route baseline. Even in the *Head-only* setting, Pre-Route consistently outperforms Self-Route, with higher QA, much lower LC usage, and higher routing accuracy. The *Generated-Meta* variant further narrows the gap to *Full-Meta*, yielding very similar QA and accuracy with comparable LC rates. *Thus, leveraging low-cost metadata is a deliberate engineering trade-off*: in realistic deployments, length and head are almost always available, other metadata can typically be derived from existing fields or light preprocessing, and the Generated-Meta fallback remains effective when such fields are missing.

Table 8: Effect of metadata settings (Head-only, Generated-Meta, Full-Meta) on routing performance on the LaRA test split.

| Answer Model | Router Model | Setting | QA↑ | LC(%)↓ | Acc↑ |
|---|---|---|---|---|---|
| DeepSeek-R1 | Pre-Route(Q235B) [N] | Head-only | 3.42 | 20.3 | 0.68 |
| | | Generated-Meta | 3.45 | 20.6 | 0.70 |
| | | Full-Meta | 3.47 | 27.2 | 0.70 |
| | Pre-Route(D-Q1.7B) [N] | Head-only | 3.43 | 18.5 | 0.69 |
| | | Generated-Meta | 3.47 | **15.8** | **0.73** |
| | | Full-Meta | **3.51** | 20.6 | **0.73** |
| | Self-Route (Baseline) | | 3.36 | 31.4 | 0.52 |
| Qwen3-235B | Pre-Route(Q235B) [N] | Head-only | 3.39 | 23.0 | 0.65 |
| | | Generated-Meta | **3.44** | 23.2 | **0.70** |
| | | Full-Meta | 3.43 | 27.2 | 0.67 |
| | Pre-Route(D-Q1.7B) [N] | Head-only | 3.40 | 21.5 | 0.66 |
| | | Generated-Meta | 3.43 | **20.4** | **0.70** |
| | | Full-Meta | 3.43 | 22.7 | 0.69 |
| | Self-Route (Baseline) | | 3.34 | 33.9 | 0.52 |

## 5 CONCLUSION

We introduced PRE-ROUTE, a proactive routing framework that activates the latent ability of LLMs to choose between retrieval-augmented generation and long-context before answering. Using lightweight structured reasoning, Pre-Route improves routing accuracy, boosts QA quality, and reduces unnecessary LC usage. Experiments on LaRA (in-domain) and LongBench-v2 (out-of-domain) show that Pre-Route outperforms Self-Route and fixed baselines, achieving better accuracy–cost trade-offs across models and configurations. Through knowledge distillation, small models inherit planning ability from larger teachers, enabling plug-and-play routing with generalization at minimal cost. These results confirm that routing ability is latent but activatable, paving the way for more efficient and interpretable long-context systems. **Future Work.** Future research may extend Pre-Route beyond binary decisions and to multi-modal and open-domain agentic scenarios, explore hybrid or sophisticated routing, and even incorporate *runtime constraints* such as budget, latency, or user preferences for adaptive deployment.

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

# 6 APPENDIX

## 6.1 RELATED WORK

**Long-context LLMs.** Improving the ability of LLMs to handle long inputs has long been a key challenge. Early work reduced computational cost by modifying attention (Beltagy et al., 2020; Guo et al., 2022) or compressing inputs (Jiang et al., 2024), while others explored distillation (Hsieh et al., 2023) and cascaded mechanisms (Chen et al., 2024). Recently, frontier models such as GPT-5 (OpenAI, 2025), Gemini-2.5 (Comanici et al., 2025), Qwen-3 (Yang et al., 2025), and DeepSeek (DeepSeek, 2025) have extended context windows to 128K tokens or way beyond. However, due to the quadratic complexity of Transformers, long-context reasoning remains expensive and unstable under ultra-long inputs, e.g., the "lost-in-the-middle" problem (Liu et al., 2024).

**Retrieval-Augmented Generation (RAG).** Another parallel line is RAG (Lewis et al., 2020)), which augments LLMs with external retrieval. RAG shows strong performance in language modeling (Khandelwal et al., 2020; Shi et al., 2024) and open-domain QA (Guu et al., 2020; Izacard & Grave, 2021), at much lower cost than modeling entire documents (Borgeaud et al., 2022). Subsequent work proposed answer correction (Yan et al., 2024), critique (Asai et al., 2024), verification (Li et al., 2024b), and adaptive retrieval (Wang et al., 2023; Cheng et al., 2024; Jeong et al., 2024), enhancing robustness in knowledge-intensive scenarios. Although retrieval introduces some overhead, RAG pipelines are modular and can be deployed locally.

**Benchmarking RAG vs. LC.** Evaluation has drawn attention as well. A number of benchmarks have been proposed, including both synthetic stress tests (Kamradt, 2023; Hsieh et al., 2024; Song et al., 2025) and task-oriented datasets (Bai et al., 2024a; An et al., 2024; Dasigi et al., 2021; s Kočiský et al., 2018; Pang et al., 2022). These studies revealed performance degradation as context length grows (Levy et al., 2024; Hsieh et al., 2024) and phenomena such as lost-in-the-middle (Liu et al., 2024). Nevertheless, most benchmarks do not exceed 128K tokens, limiting their ability to fully assess ultra-long context capabilities. This has motivated larger-scale benchmarks such as **LaRA** (Li et al., 2025), **LongBench-v2** (Bai et al., 2024b), and other related efforts (Zhang et al., 2024b; Shaham et al., 2023; Dong et al., 2024; Li et al., 2024a; Wang et al., 2024), which better match current LLM capabilities.

**The LaRA benchmark.** LaRA (Li et al., 2025) provides a more systematic comparison of RAG and LC in modern LLMs. It contains test cases across four tasks—*localization, comparison, reasoning, hallucination detection*—and three types of natural long documents: *novels, papers, reports*. Experiments on **7 open-source** and **4 proprietary** models show clear complementarity:

- **Model capability:** weaker models rely more on RAG, while stronger ones (e.g., GPT-5, Gemini 2.5) perform better with LC.

- **Task type:** LC excels in reasoning and comparison, while RAG is more robust for hallucination detection and refusals.

- **Text structure:** structured texts (e.g., reports, papers) favor LC, while novels make RAG more cost-effective.

- **Context position & retrieval granularity:** LC suffers from "lost-in-the-middle," while RAG can cross-span retrieve, but is sensitive to chunk size and number.

**Routing strategies and Self-Route.** To combine both paradigms, routing-based methods have been explored. The most representative is Self-Route (Li et al., 2024c), which adopts a *failure-driven* mechanism: the model first answers using RAG, then falls back to LC if it outputs *unanswerable*. This avoids explicit task recognition, is simple to implement, and works reasonably well in some cases. However, it has clear limitations: (1) reliance on retrieval failure signals without proactive task or information awareness; (2) extra cost from executing RAG before fallback; (3) misrouting due to over- or under-confidence; (4) lack of interpretability in decision logic. Optimizing **efficiency and performance** remains a central challenge for future routing methods.

## 6.2 OPTIMALITY OF THE IDEAL LABEL

**Proposition 1** (Optimality of the Ideal Label). *Consider the ideal label defined in Eq. 3, which prescribes LC if $U(\mathrm{LC}; q, \mathcal{D}) > U(\mathrm{RAG}; q, \mathcal{D})$ and RAG otherwise (i.e., whenever $U_{\mathrm{LC}} \leq U_{\mathrm{RAG}}$). This rule ensures optimal QA performance by construction, and among all decision rules $\pi$ that also achieve optimal performance, the ideal label minimizes the expected cost by defaulting to the lower-cost RAG in tie cases.*

*Proof.* **Step 1 (Performance).** By definition (Eq. 3), the ideal label selects the option with maximal performance: LC when $U_{\mathrm{LC}} > U_{\mathrm{RAG}}$, and RAG otherwise. Thus no alternative strategy can yield higher performance on any instance, so QA performance is optimal.

**Step 2 (Cost).** The cost of a single query can be written as

$$C_{\mathrm{answer}} = \mathbf{1}\{y = \mathrm{LC}\}\, C_{\mathrm{LC}} + \mathbf{1}\{y = \mathrm{RAG}\}\, C_{\mathrm{RAG}},$$

where $C_{\mathrm{LC}} > C_{\mathrm{RAG}} \geq 0$. Suppose an alternative strategy $\pi'$ makes the same decisions as the ideal label on strictly-better cases, but deviates on a subset $\mathcal{T}$ of tie cases by choosing LC instead of RAG. For each $i \in \mathcal{T}$, performance remains equal, but cost strictly increases by $\Delta C_i = C_{\mathrm{LC}} - C_{\mathrm{RAG}} > 0$. Hence

$$\mathbb{E}[C(\pi') - C(\pi)] = \sum_{i \in \mathcal{T}} p_i \Delta C_i \; > \; 0,$$

where $p_i$ is the probability of case $i$.

**Conclusion.** The ideal label therefore ensures optimal QA performance by construction, and among all decision rules $\pi$ that also achieve optimal performance, the ideal label minimizes the expected cost by defaulting to the lower-cost RAG in tie cases. $\square$ $\square$

## 6.3 DATASETS DETAILS

We provide additional details of the datasets used in our experiments.

Table 9: Statistics of LongBench-v2 datasets.

| Category | #data | Length |
|---|---|---|
| I. Single-Document QA | 175 | 51k–96k |
| II. Multi-Document QA | 125 | 15k–129k |
| III. Long In-context Learning | 81 | 61k–132k |
| IV. Dialogue Understanding | 39 | 25k–77k |
| V. Code Repository | 50 | 167k |
| VI. Structured Data | 33 | 49k–52k |
| **Total** | 503 | up to 167k |

Table 10: Statistics of LaRA.

| Context | Location | Reasoning | Comparison | Hallucination |
|---|---|---|---|---|
| **32k length** | | | | |
| Novel | 25673 | 25908 | 25681 | 25433 |
| Financial | 27548 | 27531 | 27546 | 27527 |
| Paper | 28078 | 28088 | 27708 | 28081 |
| # of Cases | 276 | 230 | 151 | 230 |
| **128k length** | | | | |
| Novel | 96452 | 96226 | 95903 | 96182 |
| Financial | 92684 | 92831 | 92830 | 92812 |
| Paper | 93911 | 93818 | 94731 | 93890 |
| # of Cases | 489 | 374 | 198 | 378 |

Table 11: Statistics of LaRA distribution of the test set (0.7:0.1:0.2 split).

| Context | Location | Reasoning | Comparison | Hallucination |
|---|---|---|---|---|
| **32k length** | | | | |
| Novel | 16 | 13 | 10 | 14 |
| Financial | 33 | 24 | 14 | 20 |
| Paper | 10 | 11 | 6 | 12 |
| # of Cases | 59 | 48 | 30 | 46 |
| **128k length** | | | | |
| Novel | 50 | 35 | 15 | 32 |
| Financial | 19 | 16 | 10 | 20 |
| Paper | 27 | 27 | 17 | 27 |
| # of Cases | 96 | 78 | 42 | 79 |
| **Overall (32k + 128k)** | | | | |
| Novel | 66 | 48 | 25 | 46 |
| Financial | 52 | 40 | 24 | 40 |
| Paper | 37 | 38 | 23 | 39 |
| # of Cases | 155 | 126 | 72 | 125 |

## 6.4 ADDITIONAL COMPARISON WITH ANSWER-DIRECTLY AND UNCONSTRAINED-COT BASELINES

In Sec. 2, we use the *Answer Directly* and *Unconstrained CoT* prompts as intermediate variants to probe latent routing ability. For completeness, we also evaluate these two settings as full routing strategies under the same experimental configuration as the main LaRA experiments. Tab. 12 reports QA score, LC selection rate, and routing accuracy (Acc) on the LaRA test split for all backbone models and answer configurations.

These results confirm that the conclusions in the main text are robust even if *Answer Directly* and *Unconstrained CoT* are treated as full routing baselines. While these two settings are primarily used in Sec. 2 as intermediate prompts to probe latent routing ability, Tab. 12 shows that, when evaluated as complete strategies, they still underperform Pre-Route in terms of QA, LC usage, and routing accuracy. In other words, Pre-Route improves both effectiveness and efficiency beyond these stronger prompt-based alternatives, further strengthening our main claim about its performance–cost advantages.

Table 12: Full comparison including Answer-Directly and Unconstrained-CoT prompts on the LaRA test split.

| | Qwen3-1.7B [N] | | | Qwen3-1.7B [T] | | | Qwen3-8B [N] | | | Qwen3-8B [T] | | | Qwen3-235B [N] | | | Qwen3-235B [T] | | | DeepSeek-R1 [T] | | | Qwen-Max [N] | | |
|---|---|---|---|---|---|---|---|---|---|---|---|---|---|---|---|---|---|---|---|---|---|---|---|---|
| | QA↑ | LC(%)↓ | Acc↑ | QA↑ | LC(%)↓ | Acc↑ | QA↑ | LC(%)↓ | Acc↑ | QA↑ | LC(%)↓ | Acc↑ | QA↑ | LC(%)↓ | Acc↑ | QA↑ | LC(%)↓ | Acc↑ | QA↑ | LC(%)↓ | Acc↑ | QA↑ | LC(%)↓ | Acc↑ |
| Always-LC (Baseline) | 2.13 | 100 | 0.18 | 2.29 | 100 | 0.18 | 2.98 | 100 | 0.31 | 3.23 | 100 | 0.35 | 3.46 | 100 | 0.34 | 3.51 | 100 | 0.40 | 3.44 | 100 | 0.35 | 3.36 | 100 | 0.39 |
| Always-RAG (Baseline) | 2.70 | 0 | 0.82 | 2.89 | 0 | 0.82 | 3.13 | 0 | 0.69 | 3.22 | 0 | 0.65 | 3.32 | 0 | 0.66 | 3.33 | 0 | 0.60 | 3.38 | 0 | 0.65 | 3.20 | 0 | 0.61 |
| Self-Route (Baseline) | 2.22 | 33.6 | 0.49 | 2.40 | 31.7 | 0.48 | 3.04 | 28.1 | 0.55 | 3.23 | 30.7 | 0.57 | 3.39 | 41.1 | 0.56 | 3.34 | 33.9 | 0.52 | 3.36 | 31.4 | 0.52 | 3.28 | 36.5 | 0.56 |
| Answer Directly (Q235B) [T] | 2.61 | 15.5 | 0.72 | 2.81 | 19.4 | 0.71 | 3.07 | 52.4 | 0.46 | 3.21 | 49.3 | 0.44 | 3.40 | 52.4 | 0.53 | 3.34 | 47.7 | 0.51 | 3.41 | 37.2 | 0.57 | 3.34 | 57.0 | 0.47 |
| Unconstrained CoT (Q235B) [T] | 2.63 | 17.6 | 0.71 | 2.82 | 21.7 | 0.71 | 3.08 | 51.7 | 0.49 | 3.24 | 48.7 | 0.42 | 3.33 | 50.1 | 0.49 | 3.29 | 52.4 | 0.51 | 3.40 | 33.2 | 0.59 | 3.33 | 58.5 | 0.44 |
| Answer Directly (Q1.7B) [T] | 2.34 | 58.8 | 0.43 | 2.57 | 55.1 | 0.46 | 3.05 | 67.3 | 0.41 | 3.20 | 65.4 | 0.40 | 3.33 | 59.8 | 0.43 | 3.34 | 60.0 | 0.38 | 3.39 | 64.6 | 0.45 | 3.34 | 56.7 | 0.46 |
| Unconstrained CoT (Q1.7B) [T] | 2.43 | 37.2 | 0.57 | 2.61 | 39.4 | 0.56 | 2.99 | 72.3 | 0.39 | 3.21 | 70.6 | 0.41 | 3.34 | 67.3 | 0.36 | 3.35 | 70.3 | 0.37 | 3.37 | 64.9 | 0.39 | 3.35 | 69.1 | 0.42 |
| Pre-Route (R1) [T] | 2.70 | 2.7 | 0.83 | 2.88 | 1.8 | 0.83 | 3.15 | 5.0 | 0.76 | 3.23 | 8.0 | 0.70 | 3.39 | 14.2 | 0.73 | 3.37 | 10.8 | 0.68 | 3.42 | 10.0 | 0.70 | 3.24 | 10.6 | 0.67 |
| Pre-Route (Q235B) [N] | 2.71 | 4.7 | 0.83 | 2.90 | 4.8 | 0.82 | 3.16 | 22.2 | 0.73 | 3.30 | 26.2 | 0.70 | 3.47 | 29.1 | 0.72 | 3.43 | 27.2 | 0.67 | 3.47 | 27.2 | 0.70 | 3.31 | 23.1 | 0.69 |
| Pre-Route (Q235B) [T] | 2.71 | 2.3 | 0.84 | 2.91 | 3.0 | 0.84 | 3.17 | 15.9 | 0.74 | 3.27 | 17.9 | 0.71 | 3.42 | 20.2 | 0.72 | 3.40 | 18.3 | 0.68 | 3.45 | 16.9 | 0.71 | 3.29 | 19.6 | 0.69 |
| Pre-Route (Q1.7B) [N] | 2.70 | 7.9 | 0.79 | 2.85 | 10.8 | 0.77 | 3.12 | 34.0 | 0.60 | 3.29 | 33.2 | 0.63 | 3.36 | 40.9 | 0.53 | 3.41 | 40.6 | 0.54 | 3.44 | 32.5 | 0.56 | 3.25 | 36.6 | 0.55 |
| Pre-Route (Q1.7B) [T] | 2.68 | 10.8 | 0.77 | 2.87 | 11.1 | 0.80 | 3.10 | 35.6 | 0.61 | 3.25 | 39.9 | 0.58 | 3.41 | 35.6 | 0.59 | 3.41 | 33.1 | 0.59 | 3.49 | 31.5 | 0.68 | 3.26 | 28.3 | 0.60 |
| Pre-Route (D-Q1.7B) [N] | 2.70 | 3.6 | 0.82 | 2.89 | 3.9 | 0.83 | 3.16 | 21.5 | 0.73 | 3.30 | 21.4 | 0.71 | 3.46 | 24.6 | 0.74 | 3.43 | 22.7 | 0.69 | 3.51 | 20.6 | 0.73 | 3.28 | 24.7 | 0.67 |
| Pre-Route (D-Q1.7B) [T] | 2.71 | 3.2 | 0.83 | 2.89 | 2.8 | 0.82 | 3.19 | 19.5 | 0.76 | 3.28 | 20.8 | 0.70 | 3.47 | 26.9 | 0.73 | 3.44 | 26.0 | 0.67 | 3.47 | 20.3 | 0.72 | 3.31 | 24.5 | 0.69 |

## 6.5 CASE STUDY ON THE NECESSITY OF A FINER-GRAINED EVALUATION METRIC

To illustrate why binary evaluation is insufficient, we present a case study highlighting the value of a 4-point scoring rubric. This example shows how two answers may both be judged "correct" under binary metrics, yet differ substantially in completeness and alignment with ground truth. The case study below (Tab. 6.5) concretely demonstrates how finer-grained evaluation better reflects qualitative differences.

---

**Case Study: Analysis of the "Stella and Miles" Reasoning Query**

1) INITIAL ANALYSIS: THE QUERY AND RAG'S SCOPE

**Query:** *"Why does Stella repeatedly think that their father will 'kill' Miles?"*

The presence of the keyword **repeatedly** is crucial, as it implies that the necessary context is likely scattered across multiple passages, requiring a broader understanding than localized snippets might provide.

**RAG Retrieved Snippets Analysis:** The retrieved passages contain phrases like **"Miles was expelled from school"** and **"dad will be very angry."** These snippets allow a model to construct a direct causal chain **but inherently lack the historical context needed to address the "repeatedly" aspect of the query**.

**Ground Truth:** Stella knows that Miles has been expelled from multiple schools, and when she realizes he got kicked out again, she's convinced their father will be furious and punish Miles harshly because it's an ongoing issue, and they have high expectations for him.

2) COMPARATIVE ANALYSIS AND SCORING

**Long-Context Answer (Score: 4/4):** "Stella repeatedly thinks their father will 'kill' Miles because she knows he was expelled **again**. She anticipates not only her father's anger but also his **disappointment**, given the **high expectations** they have for Miles, and thus fears a **severe outcome**."

**RAG Answer (Score: 3/4):** "Stella thinks their father will 'kill' Miles because Miles was expelled from school, and she knows their father will be very angry and punish him severely."

**Analysis:** Under a binary (correct/incorrect) system, both answers would likely be marked "correct" as they identify the core reason (the expulsion). However, this masks a substantial quality gap. To properly evaluate them, we compare them against the key components of the ground truth:

- The expulsion is a repeated event (**again / multiple schools**).
- The father has **high expectations** for Miles.
- The father's reaction includes extreme **anger and disappointment**.
- The consequences are expected to be **severe**.

The **RAG answer** covers the basic causal chain but fails to address the crucial contexts of **repetition** and **high expectations**. In contrast, the **Long-Context answer** successfully incorporates these nuances, aligning almost perfectly with the ground truth.

Our 4-point rubric is designed to capture these critical distinctions. By assigning a higher score to the Long-Context answer, we can accurately reflect its deeper contextual alignment and superior quality.

**Conclusion:** The Long-Context strategy is demonstrably superior for this query (`better_strategy = long_context`).

---

## 6.6 CASE STUDIES ON WHY SELF-ROUTE SOMETIMES FAILS ON LARA

Across our case studies, a consistent pattern emerges: **Self-Route tends to fail at the extremes of confidence**.

- **Over-confidence.** When retrieved chunks appear superficially relevant, Self-Route prematurely concludes that RAG suffices, even if the query requires *distributed reasoning* across multiple parts of the document. In Case 1, it ignored the global cue "repeatedly" and thus failed to capture the need for long-context integration.

- **Under-confidence.** Conversely, when the evidence is present but takes the form of a *negative premise check* ("not discussed") or a *keyword-localized fact* (e.g., a single time point), Self-Route misclassifies the situation as "unanswerable." This leads to unnecessary escalation to long-context models, wasting cost and providing no additional benefit (Cases 2 and 3).

In other words, Self-Route lacks a calibrated notion of when retrieved evidence is *sufficient* versus when it is *incomplete*. Our router, by relying on lightweight meta signals (question form, genre, and evidence distribution cues), avoids these extremes: it dispatches *distributed* queries to Long-Context (Case 1), while keeping *localized* or *premise-check* queries on RAG (Cases 2 and 3).

---

**Case Study 1: "Why does Stella repeatedly think their father will 'kill' Miles?"**

**What was asked?**
Explain *why* Stella keeps thinking this (note the word *"repeatedly"*).

**Where is the evidence?**
Scattered across the story: repeated expulsions (*again*), strict family expectations, and the fear of harsh punishment.

**What did the two answering strategies say?**

- **RAG (picked by Self-Route):** Gave the short cause: Miles was expelled ⇒ father angry/punish. *Missed* the "*again*" and "high expectations". **Score: 3**.

- **Long-Context (picked by our router):** Added the missing parts: expelled *again*, family expectations, and "angry + *disappointed*". **Score: 4**.

**Who routed what? Why?**

- **Self-Route** chose **RAG** because retrieved chunks looked answerable.

- **Our router** chose **Long-Context** because *"repeatedly"* means the proof is spread out.

**Why did Self-Route miss?**
It treated "retrieved chunks can answer" as enough and ignored the global signal *"repeatedly"*.

**Why our routing was correct.** Using meta signals only: the question includes *"repeatedly"*, the genre is long-form narrative, and the document fits the window. These cues imply evidence is *distributed* across scenes, so **Long-Context** is the appropriate choice.

---

### Case Study 2: "Did the paper discuss ReLU-at-classification for NLP?"

**What was asked?**
Check whether *paper 1* talks about using ReLU as a *classification layer* for *NLP*, compared with traditional models.
**Where is the evidence?**
Right in the abstract/introduction: the paper runs on **MNIST, FashionMNIST, WDBC** (image/biomedical), *not NLP*.
**Ground truth.**
"*Paper 1 does not address the impact . . . on NLP tasks.*"
**What did the two answering strategies say?**

- **RAG:** Said the paper *does not discuss* NLP (correct). **Score: 4**.

- **Long-Context:** Same conclusion (also correct). **Score: 4**.

**Who routed what? Why?**

- **Self-Route** sent it to **Long-Context** after treating checking retrieved chunks and made"not discussed" as "unanswerable".

- **Our router** picked **RAG** because this is a *premise check*; the abstract is enough.

**Why did Self-Route miss?**
It confused a valid negative answer (*"not discussed"*) with "unanswerable", and used more context than needed.
**Why our routing was correct.** Based solely on meta information: the query is a *premise check* ("did it discuss . . . ?"), and such evidence is typically *localized* in the abstract/introduction. Under these cues and the "prefer RAG on ties" rule, **RAG** is recommended for efficiency.

### Case Study 3: "On April 2 when did the whole view fall into darkness?"

**What was asked?**
A single time point on April 2.
**Where is the evidence? (verbatim)**
". . . *until, **at five minutes before seven**, the whole surface in view was enveloped in the darkness of night.*"
**What did the two answering strategies say?**

- **RAG:** "*at five minutes before seven (6:55)*" **Score: 4**.

- **Long-Context:** "*At five minutes before seven.*" **Score: 4**.

**Who routed what? Why?**

- **Self-Route** escalated to **Long-Context** after saying "unanswerable".

- **Our router** chose **RAG** because this is a *keyword-localizable* fact (date/event/time).

**Why did Self-Route miss?**
Even though the answer is in retrieved chunks, somehow Self-Route failed to give answer.
**Why our routing was correct.** From meta-only cues: the query asks for a *single time point* with clear date/time keywords that usually appears in *one sentence*. These signals favor **RAG** for precise, keyword-localizable extraction.

## 6.7 EXAMPLE PROMPTS

We provide the full prompts used in our experiments. Figs. 5 and 6 shows the original Self-Route prompt, while Figs 7 and 8 present evaluation prompts adapted from LaRA for hallucination detection and general answer comparison. Finally, Fig. 9 gives the complete Pre-Route prompt, which integrates structured reasoning steps and decision rules.

---

**Complete Prompt for Self-Route**

```
Answer the question based on the given passages. Only give me the answer and do not
    ↪ output any other words. If the question cannot be answered based on the
    ↪ information in the article, write 'unanswerable'. The following are given
    ↪ passages. {rag_result} Question: {query} Answer:
```

---

Figure 5: The complete Self-Route Prompt for LaRA. Taken directly from its original paper (Li et al., 2024c)

---

**Complete Prompt for Self-Route**

```
Read the following text and answer briefly. {rag_chunks} Now, answer the following
    ↪ question based on the above text, only give me the answer and do not output any
    ↪  other words. If the question cannot be answered based on the information in
    ↪ the article, write 'unanswerable'. Question: {query} Answer:
```

---

Figure 6: The complete Self-Route Prompt for Longbench-v2 multiple-choice. Taken directly from its original paper (Li et al., 2024c)

```
Evaluation Prompt for Hallucination Detection (Adapted from LaRA)

You will be given a question, a groundtruth answer, and two answers from AI assistants
    ↪ (Assistant A and Assistant B).
This question may not be directly answerable from the source text.

Your task:
1. Check whether each assistant is hallucinating. If the answer is **consistent** with
    ↪ the groundtruth, and the question is **not present in the original text**, it
    ↪ is considered **not hallucinating**.
2. For each answer, assign a quality score from 1 to 4:
    - 4 = Fully correct (not hallucinating and consistent)
    - 3 = Mostly correct (minor detail missing or uncertain)
    - 2 = Partially correct (some correct points, some hallucination)
    - 1 = Hallucinated, irrelevant, or incorrect
3. Write a brief comparative analysis.
4. Decide who is better: "A", "B", or "Tie"

Return JSON in this format:
{{
  "analysis": "...",
  "score_a": int,
  "score_b": int,
  "better": "A" | "B" | "Tie"
}}

[Question]
{query}

[Groundtruth Answer]
{label}

[Assistant A's Answer]
{pred_a}

[Assistant B's Answer]
{pred_b}

Start your evaluation:
```

Figure 7: An adapted prompt from the LaRA benchmark, specifically refined for evaluating hallucination. It provides a detailed scoring rubric to distinguish between factual consistency and fabrication.

**Standard Evaluation Prompt for Answer Comparison (Adapted from LaRA)**

```
You will be given a question, a groundtruth answer, and two AI assistant answers.

Your task:
1. Judge each answer's factual accuracy and completeness relative to the groundtruth.
2. For each answer, assign a score from 1 to 4:
   - 4 = Fully correct (covers all key points, no factual or logical flaws)
   - 3 = Mostly correct (core is right, minor flaws or omissions)
   - 2 = Partially correct (some accurate parts, but notable errors or gaps)
   - 1 = Incorrect or hallucinated (serious flaws, irrelevant, fabricated)
3. Provide a brief comparison explaining your scoring.
4. Choose which assistant is better: "A", "B", or "Tie"

Return your result in the following JSON format:
{{
  "analysis": "...",
  "score_a": int,
  "score_b": int,
  "better": "A" | "B" | "Tie"
}}

[Question]
{query}

[Groundtruth Answer]
{label}

[Assistant A's Answer]
{pred_a}

[Assistant B's Answer]
{pred_b}

Start your evaluation:
```

Figure 8: A general-purpose evaluation prompt adapted from LaRA for detailed comparison. This version focuses on factual accuracy, completeness, and overall quality against a groundtruth answer.

1242
1243
1244
1245
1246
1247
1248
1249
1250
1251
1252
1253
1254
1255
1256
1257
1258
1259
1260
1261
1262
1263
1264
1265
1266
1267
1268
1269
1270
1271
1272
1273
1274
1275
1276
1277
1278
1279
1280
1281
1282
1283
1284
1285
1286
1287
1288
1289
1290
1291
1292
1293
1294
1295

---

**Complete Prompt for Pre-Route**

```
## Inputs
- query: "{query}"
- task_type: {task_type}
- document_title: "{doc_title}"
- document_type: {doc_type}
- document_length: {length_str} ({doc_len_tokens} tokens)
- answering_model: {model}
- answering_max_window: {max_window_tokens} tokens
- document_fits_window: {document_fits}
- document_head content: "{doc_head}..."
## RAG Configuration
- chunk_size: {chunk_size} tokens
- chunk_overlap: {chunk_overlap} tokens
- embed_model: {embed_model}
- rerank_model: {rerank_model}
- vector_ratio: {vector_ratio}
- rerank_size: {rerank_size}
## Instructions
You are tasked with choosing the most appropriate strategy — **RAG**, **LONG_CONTEXT**
    ↪ — for answering the user query, based on the characteristics of the query,
    ↪ document, and model.
Please complete the `<thinking>` block below. In each step, provide a clear judgment
    ↪ and explain how it affects your strategy choice.
1. `<step1>`: Identify the question type (e.g., factual, reasoning, comparison,
    ↪ judgment, etc.) and the document type (e.g., book, article, report). How do
    ↪ these affect the need for deep context understanding or precise retrieval?
2. `<step2>`: Assess whether the relevant information is likely concentrated in one
    ↪ part of the document or scattered across multiple sections. How does this
    ↪ affect strategy selection?
3. `<step3>`: Evaluate whether the document can fully fit into the context window,
    ↪ based on `document_fits_window` above. If not, how does that impact strategy
    ↪ choice?

4. `<step4>`: Consider whether the query can be answered through keyword-based
    ↪ retrieval (e.g., names, dates), or requires synthesizing implicit logic,
    ↪ analogies, or multi-part reasoning.
5. `<step5>`: Reflect on the model being used (e.g., {model}). Consider both its
    ↪ context window size and model capacity (parameters). Although some models may
    ↪ have large context windows, smaller models may still struggle with effective
    ↪ long-context reasoning due to limited capacity. How does this influence your
    ↪ strategy recommendation?
6. `<step6_efficiency>`: Compare **expected efficiency** of RAG vs LONG_CONTEXT:
    ↪ expected context size, retrieval selectivity, latency, and cost. If quality is
    ↪ likely similar, which strategy is more efficient?
7. `<reflection>`: Based on your reasoning above, state which strategy is more suitable
    ↪ overall — **RAG**, **LONG_CONTEXT** — and explain why.
8. `<decision>`: Write your final strategy choice clearly as either `RAG`, `
    ↪ LONG_CONTEXT`.
### Decision Rules
- If both strategies are **equally suitable** or quality difference is **negligible/
    ↪ uncertain**, **prefer RAG for efficiency**.
- Prefer **LONG_CONTEXT** only if (a) the document **fits** in the window **and** (b)
    ↪ the query requires **global, cross-section synthesis** that retrieval would
    ↪ likely miss.
- Prefer **RAG** when the document **does not fit** the window, or when **high-
    ↪ precision snippet retrieval** is likely effective (e.g., names, dates,
    ↪ localized facts), or when **efficiency** is a concern and quality is similar.
## Output Format
<thinking>
  <step1>...</step1>
  <step2>...</step2>
  <step3>...</step3>
  <step4>...</step4>
  <step5>...</step5>
  <step6_efficiency>...</step6_efficiency>
  <reflection>...</reflection>
  <decision>...</decision>
</thinking>
```

Figure 9: The complete Pre-Route Prompt.

## 6.8 ETHICS STATEMENT

This work does not involve human subjects, sensitive data, or applications that may raise ethical concerns. Our experiments are conducted on publicly available datasets, and we adhere to the ICLR Code of Ethics throughout this research.

## 6.9 REPRODUCIBILITY STATEMENT

We have made efforts to ensure reproducibility. The main paper describes the proposed method, model configurations, and evaluation protocols. Additional training details, hyperparameters, and extended results are provided in the appendix. A code repository is also provided[1] to facilitate reproducibility.

---

[1]https://anonymous.4open.science/r/pre-route-FCE5

