# OpenReview forum: "Route Before Retrieve: Activating Latent Routing Abilities of LLMs for RAG vs. Long Context Selection"
_ICLR.cc/2026/Conference — Submitted to ICLR 2026_

### Official Review · Reviewer_bHjd · 2025-10-28

**Soundness:** 2
**Presentation:** 2
**Contribution:** 3
**Rating:** 4
**Confidence:** 3

**Summary:**

This paper investigates how to choose an optimal strategy over **Long Context** and **RAG** for long context LLM situations. The methods use LLMs to decide the strategy based on the meta information of the long context. The experiments over LARK and LongBenchv2 showed that the proposed method achieves lower cost and better performance.

**Strengths:**

1. The studied problem is important, as it is useful for long context LLM inference.
2. The method is novel, where LLMs utilize meta information to judge the strategy. The results on two classic Benchmark is validated as effective.

**Weaknesses:**

1. The presentation is hard to understand, especially for the methods. Do you explain what the meaning of (m_i, T_i, y_i) is in equation 4?
2. The accuracy of methods over the benchmarks is very low. I know it's not caused by methods. However, it's weird to see the analysis of such poor performance.
3. The metadata is not available for every long context situation. Do you provide  the solution for such problems?

**Questions:**

See weakness.

---

> ### Author Response · Authors · 2025-11-17
> **Response to Weakness 1&2**
>
> ### W1 Response
> We thank the reviewer for pointing out the clarity issue regarding Equation (4). We agree that the current version provides only a brief explanation of the notation, which may cause confusion.
>
> In our formulation, the triplet $(m_i, T_i, y_i)$ in Equation (4) has the following meaning:
>
> - **$m_i$**: the *metadata* of the $i$-th example, including document title, type, length, and the initial text segment;
> - **$T_i$**: the *reasoning trace* (thought chain) generated by the teacher model $\pi_T$ before producing a routing decision;
> - **$y_i$**: the *routing decision* made by the teacher model (RAG or LC);
>    and $\hat{y}_{ideal,i}$ denotes the *ideal label* defined from QA scores.
>
> Thus, Equation (4) specifies the filtering stage in **Stage 1 (Rejection Sampling)**: we retain only those examples where the teacher’s decision matches the ideal label, ensuring high-quality supervision signals. We already clarify these notations in the revised version to improve readability and consistency.
>
> ### W2 Response
> We understand that, when looking at the scores in the table, the reviewer may have the intuitive impression that the overall scores are quite low. We assume you are mainly referring to the results on **LongBench-v2** (please kindly correct us if this is not the case). We would like to clarify that **this is due to the evaluation scale and difficulty of this benchmark itself, rather than a failure specific to our method.**
>
> In *LongBench v2: Towards Deeper Understanding and Reasoning on Realistic Long-context Multitasks* (arXiv:2412.15204, https://arxiv.org/pdf/2412.15204), its Table 2 reports the overall performance (the “Overall” column) as follows:
>
> - Human expert (Human*): 53.7%
> - Proprietary models: GPT-4o-2024-08-06 50.1%, o1-preview-2024-09-12 57.7%, Claude-3.5-Sonnet 41.0%
> - Open-source models: Qwen2.5-72B-Instruct 39.4%, Llama-3.1-70B-Instruct 31.6%, Qwen2.5-7B-Instruct 27.0%
>
> As we can see, even human experts and strong proprietary models only achieve around 40–60% overall accuracy on LongBench-v2, while most open-source models fall in the 30–40% range. Therefore, the fact that the absolute scores “look low” is actually **a property of the LongBench-v2 benchmark itself, rather than an anomaly of our approach.**
>
> Under this evaluation scale, the Overall scores of the models in our Table 6 are consistent with the ranges reported for comparable models in the LongBench-v2 paper, indicating that our results are **normal and reasonable**. Our goal in using LongBench-v2 is to treat it as a challenging OOD scenario and examine whether Pre-Route can still effectively control LC usage while maintaining or improving QA and routing accuracy when task formats and distributions shift.

---

> ### Author Response · Authors · 2025-11-17
> **Response to Weakness 3**
>
> ### W3 Response
> We appreciate your concern. While metadata quality may vary, Pre-Route is not built on the assumption of clean or fully available metadata. It relies on **low-cost structural signals that are naturally present in most real-world systems**, and we provide explicit fallback mechanisms when such information is incomplete.
>
> First, in practical RAG and long-context pipelines, document length and the opening snippet are always available, and items such as titles or document types are typically auto-generated from filenames, URLs, or database fields. Real-world documents—webpages, chat logs, code repos, enterprise files—naturally carry structural cues (titles, timestamps, speaker IDs, file paths). These are *system-level byproducts*, not extra assumptions, making reliance on such metadata a realistic engineering choice rather than a fragile dependency.
>
> Second, even when metadata is noisy (e.g., a misleading doc_head), Pre-Route does **not** hinge on any single feature. It integrates query cues, document structure, and information-distribution reasoning. For example, in the Stella case from the appendix 6.6, despite head snippet looks like a local factual snippet of a story, cues like *“repeatedly”* and *“throughout the story”* lead Pre-Route to identify a globally distributed answer, correctly favoring LC. This shows that doc_head acts only as a soft prior, not a determinant.
>
> Third, for cases where metadata is genuinely missing (e.g., webpages or chat logs without clear doc_type), we use a lightweight **Generated-Meta** fallback: a small model produces a pseudo title/type from the query and doc_head, after which routing proceeds normally. Its overhead is minimal and remains far below Self-Route. Experiments show:
>
> - **Head-only** incurs some degradation but is still consistently stronger than Self-Route;
> - **Generated-Meta** nearly closes the gap to **Full-Meta**;
> - Across settings, both the large and distilled Pre-Route variants maintain higher QA and lower LC usage than Self-Route.
>
> Thus, leveraging low-cost metadata is a deliberate engineering choice, not a weakness of the method. In real scenarios, the document head and length are virtually always available, and most other metadata can be extracted from existing fields or simple preprocessing. For the rare cases where metadata is absent, the Generated-Meta fallback provides a reliable solution backed by empirical results. With these mechanisms, Pre-Route remains robust even on “messier,” less structured real-world data, rather than only on idealized well-formed documents.
>
> We **additionally include this discussion and the corresponding experimental results as a new subsection in Section 4 of the revised manuscript.**
>
>
> | Answer Model | Router Model           | Exp            | QA↑  | LC(%)↓ | Acc↑ |
> | ------------ | ---------------------- | -------------- | ---- | ------ | ---- |
> | DeepSeek-R1  | Pre-Route(Q235B) [N]   | Head-only      | 3.42 | 20.3   | 0.68 |
> |              |                        | Generated-Meta | 3.45 | 20.6   | 0.7  |
> |              |                        | Full-Meta      | 3.47 | 27.2   | 0.7  |
> |              | Pre-Route(D-Q1.7B) [N] | Head-only      | 3.43 | 18.5   | 0.69 |
> |              |                        | Generated-Meta | 3.47 | 15.8   | 0.73 |
> |              |                        | Full-Meta      | 3.51 | 20.6   | 0.73 |
> |              | Self-Route(Baseline)   |                | 3.36 | 31.4   | 0.52 |
> | Qwen3-235B   | Pre-Route(Q235B) [N]   | Head-only      | 3.39 | 23     | 0.65 |
> |              |                        | Generated-Meta | 3.44 | 23.2   | 0.7  |
> |              |                        | Full-Meta      | 3.43 | 27.2   | 0.67 |
> |              | Pre-Route(D-Q1.7B) [N] | Head-only      | 3.4  | 21.5   | 0.66 |
> |              |                        | Generated-Meta | 3.43 | 20.4   | 0.7  |
> |              |                        | Full-Meta      | 3.43 | 22.7   | 0.69 |
> |              | Self-Route(Baseline)   |                | 3.34 | 33.9   | 0.52 |

---

> ### Comment · Reviewer_bHjd · 2025-11-17
> **Unable to see the response.**
>
> Although I received the email about the response, I'm still unable to see the response in the OpenReview. It's very strange.

---

> > ### Author Response · Authors · 2025-11-17
> >
> > Thank you for letting us know, and we sincerely apologize for the confusion.
> > It was caused by an incorrect visibility setting when we saved the draft response. We have now fixed the setting, and the full author response should be visible on OpenReview.

---

> ### Author Response · Authors · 2025-11-26
>
> Dear Reviewer bHjd,
>
> We appreciate your valuable comments on our submission.
>
> With the discussion phase approaching its final week, we kindly inquire if we have effectively addressed your questions. If there are any remaining areas requiring further clarification, please do not hesitate to reach out.
>
> If you are satisfied with our clarifications, we would greatly appreciate your consideration in updating the evaluation score accordingly.
>
> We sincerely look forward to your feedback.
>
> Best regards, The Authors

---

> > ### Comment · Reviewer_bHjd · 2025-11-26
> >
> > Thank you for the detailed response from the authors.
> >
> > However, my concerns about the performance and relying on the metadata still remain. Thus I prefer to keep my score.

---

> > > ### Author Response · Authors · 2025-11-27
> > > **Clarification on Evaluation Context and System Robustness**
> > >
> > > Thank you for your response. We believe there may be a misalignment in expectations regarding the difficulty of the benchmark and the practical design of the system. We offer the following clarifications to provide the necessary context:
> > >
> > > **1. Addressed Benchmark's Performance Concerns:** We clarified that the lower absolute scores on LongBench-v2 are a property of the benchmark's difficulty rather than a method flaw. As referenced from the LongBench-v2 paper, **even Human Experts only achieve ~53.7% accuracy**. Our results fall within a reasonable range consistent with this challenging baseline, and we maintain a relative advantage over Self-Route.
> > >
> > > **2. Validated Metadata Robustness:** We emphasized that most metadata Pre-Route requires are **natural system-level byproducts** in real-world pipelines, not additional assumptions.
> > >
> > > - **Robustness to Noise:** Even when metadata is misleading (e.g., the "Stella" case discussed in our response), the model correctly prioritizes query cues to make the right decision.
> > > - **Fallback Mechanism:** For rare cases where metadata is absent, we introduced **Generated-Meta**. New experiments show that even with Head-only or Generated-Meta, Pre-Route **consistently outperforms the Self-Route baseline**.
> > >
> > > **Inquiry:** Regarding your remaining concerns over performance, could you kindly specify which aspect you find concerning? We would greatly appreciate the opportunity to resolve this specific doubt.

---

### Official Review · Reviewer_5nQ3 · 2025-10-28

**Soundness:** 2
**Presentation:** 2
**Contribution:** 2
**Rating:** 4
**Confidence:** 4

**Summary:**

This paper proposes Pre-Route, a framework for choosing between RAG and LC processing before answering queries . The core approach involves using lightweight metadata (document type, length, title, initial snippet) to perform structured reasoning. The authors claim LLMs possess "latent routing ability" that can be activated through structured prompts, validated via Best-of-N sampling and linear probing experiments . They distill this capability from large models (Qwen3-235B, DeepSeek-R1) into smaller ones (Qwen3-1.7B) using rejection sampling on an "ideal label" defined by QA performance with RAG preference in ties. Experiments on LaRA (in-domain) and LongBench-v2 (out-of-domain) show Pre-Route achieves better accuracy-cost trade-offs than Self-Route and fixed baselines.

**Strengths:**

**1. Comprehensive Experimental Validation**
The paper presents extensive experiments across multiple dimensions: behavioral analysis via Best-of-N sampling showing accuracy improvement from 0.53 (N=1) to 0.87 (N=8) under direct prompting; representation analysis using linear probes demonstrating that structured prompts improve ideal-label accuracy from 0.396 to 0.625 on Qwen3-1.7B; evaluations on 6+ answer models ranging from 1.7B to 235B parameters; and robustness tests under different retrieval configurations (rerank size 5/7/10)

**2. Interpretable Decision Process**
The structured six-step reasoning chain provides transparency in routing decisions, addressing a key limitation of Self-Route's black-box nature. The ablation study confirms each step contributes to the final performance, with removing decision rules causing LC rate to spike from 20.7% to 45.3

**Weaknesses:**

**1. Lack of Justification for Design Choices**
The design lacks theoretical grounding and empirical support for why these specific meta-information features and guidelines were chosen. The paper does not explain the principled basis for selecting document type, length, title, and initial snippet as the metadata inputs, nor does it justify why the six reasoning steps are structured in this particular order and formulation. No ablation study explores alternative metadata configurations or reasoning structures, making the design appear arbitrary rather than well-motivated.

**2. Highly Engineered Approach with Limited Innovation**
The method is a highly customized heuristic system—essentially a hand-crafted prompt engineering solution. The six reasoning steps (task & document characterization, distribution pattern judgment, context-window feasibility, retrieval feasibility, model capability consideration, efficiency trade-off) are manually designed heuristics that require pre-specification rather than being automatically learned or generated. This pre-designed nature significantly limits innovation, as the approach lacks unique architectural contributions beyond careful prompt construction. The core claim of "activating latent routing ability" essentially reframes better prompting as a discovery of hidden capabilities, which overstates the conceptual novelty

**3. High Inference Cost and Missing Comparison with Modern Agentic RAG Paradigms**
The framework incurs substantial additional computational overhead by requiring deployment of a separate routing LLM just to decide between RAG and LC. This stands in contrast to mainstream Agentic RAG [1][2] approaches, which internalize the routing capability within the answering model itself through reinforcement learning, enabling the model to dynamically switch between RAG and LC during generation without external routing overhead. The paper completely lacks discussion of such RL-based integrated approaches, which further diminishes its claimed innovation.

**4. Limited Generalization and Marginal Performance Gains**
The effectiveness is only substantial on in-domain data. The LongBench-v2 out-of-domain results show Pre-Route's advantage narrows considerably, with Self-Route becoming "more competitive in QA scores", suggesting the approach may not generalize beyond LaRA's 4-point scoring regime. This raises questions about whether the added complexity of distillation and separate router deployment justifies the marginal gains.

[1] Singh A, Ehtesham A, Kumar S, et al. Agentic retrieval-augmented generation: A survey on agentic rag[J]. arXiv preprint arXiv:2501.09136, 2025.

[2] Liang J, Su G, Lin H, et al. Reasoning RAG via System 1 or System 2: A Survey on Reasoning Agentic Retrieval-Augmented Generation for Industry Challenges[J]. arXiv preprint arXiv:2506.10408, 2025.

**Questions:**

**1. Missing Detailed Linear Probing Experimental Setup**
What is the exact training data and label generation procedure for the linear probes? The paper states that probes are trained on "frozen penultimate last token embeddings" , but does not specify: (a) the size of the training set used for probe fitting; (b) whether the same train/val/test split from LaRA is used or a separate split; (c) the optimization procedure (optimizer, learning rate, number of epochs); (d) how the "ideal," "route," "doc_type," and "task_type" labels are generated for the probing dataset. Without these details, the linear probing results in Table 1 cannot be reproduced or properly interpreted.

**2. Inadequate Explanation of Table 2 (Cost Analysis)**
Table 2 lacks detailed explanation of the experimental setup. Specifically: (a) What is the exact experimental configuration—are these costs averaged over the entire LaRA test set or based on representative samples? (b) Why do the two models with vastly different parameter counts (Qwen3-235B vs. Qwen3-1.7B) have identical input and output token counts (1205 input, 648 output) for Pre-Route? This seems implausible unless the prompts and reasoning chains are exactly the same length regardless of model size. (c) Why does Self-Route have 2600 input tokens compared to Pre-Route's 1205—where does this additional input come from? Is Self-Route including retrieved chunks in the routing decision, while Pre-Route uses only metadata? This critical distinction is not clarified.

**3. Potential Annotation Error in Main Results**
In Table 3 (LaRA main results), under the Qwen3-235B [T] answer model configuration, Always-LC (Baseline) achieves a QA score of 3.51, which appears to be the highest score in that column.

---

> ### Author Response · Authors · 2025-11-17
> **Response to Weakness 1&2**
>
> ### W1 Response
>
> We thank the reviewer for the thoughtful comments. We clarify that the metadata choices and the six-step reasoning structure in Pre-Route are not arbitrarily designed. Instead, they result from a systematic process grounded in decision-theoretic principles, empirical observations, and task-distribution analysis of prior studies in RAG vs. LC (e.g., [1–3]).
>
> **(1) Motivation from theory and empirical analysis**
>  The selected metadata fields (document type, length, title, document head) align with commonly recognized factors that influence the RAG–LC trade-off:
>
> - **Document type and Task type** reflects semantic structure (factual vs. narrative), which strongly affects retrieval feasibility;
> - **Document length**, together with **context-window feasibility**, determines information coverage and computational cost;
> - **Title** and **document head** provide low-cost priors about topic and information distribution.
>
> These variables were **not selected arbitrarily**, but were motivated by patterns and statistical analyse observed in prior studies. People observed distributional differences: across genres (e.g., reports, papers, novels) and length, the optimal choice between RAG and LC differs systematically. This makes the above metadata effective predictors of whether the task requires global coverage or can rely on local retrieval.
>
> **(2) Rationale for the six-step structure and its ordering**
>  The six-step reasoning structure follows a decision hierarchy from *task understanding → information distribution → feasibility → efficiency trade-off*, aligning with the decision-theoretic formulation in Eqs. (1–3). The ordering follows a natural progression—from task understanding to feasibility assessment—that we found to produce stable and interpretable reasoning.
>
> We also conducted **step-wise ablation studies** in the paper. The results show that all six steps contribute meaningfully: removing any step (e.g., information-distribution judgment or model-capability consideration) reduces routing accuracy and weakens interpretability.
>
> Thus, the design of Pre-Route **is not a random design**, but a systematic framework supported by prior analyses, empirical evidence, and step-level ablation validation, providing a coherent structure that is empirically validated through step-level ablations..
>
> [1] Z. Li et al., “Retrieval augmented generation or long-context LLMs? A comprehensive study and hybrid approach,” EMNLP 2024 (Industry Track).
>
>  [2] P. Xu et al., “Retrieval meets long context large language models,” ICLR 2024.
>
>  [3] K. Li et al., “LaRA: Benchmarking retrieval-augmented generation and long-context LLMs – no silver bullet for LC or RAG routing,” ICML 2025.
>
> ### W2 Response
>
> We thank the reviewer for the comment, but we respectfully do not fully agree with this assessment. While Pre-Route does use a structured prompt, the main contribution is **not** the template itself, but the **systematic identification and stabilization of a latent RAG–LC routing ability in LLMs**.
>
> 1. **Not pure heuristics rules, but making existing abilities explicit.**
>    The dimensions we use (task/document characterization, information distribution, etc.) are a factorization of the key variables in the RAG–LC accuracy–cost trade-off. We only make this decision space explicit in natural language; the actual boundary of *when to choose RAG vs. LC* is determined by the model’s pretrained internal knowledge, not by hard-coded heuristics.
> 2. **From prompt to learnable router.**
>    The structured reasoning is not the endpoint: we distill it into a **lightweight learned router**, showing that this behavior can be compactly learned and generalized by a small model rather than remaining a hand-crafted prompt.
> 3. **Evidence that “latent routing” is real, not rhetorical.**
>    Best-of-N sampling shows that the model already has a **strong but unstable** routing capability, and linear probes (Table 1) reveal that this signal becomes more separable in representation space under our prompting, suggesting that the model carries a substantive underlying capability, not merely a trivial heuristic..
> 4. **Beyond “one more prompt”: a practical routing framework.**
>    Pre-Route (i) upgrades RAG vs. LC from a passive fallback to an explicit **plan-then-execute** decision step, (ii) transfers this routing ability to a small, deployable router, and (iii) consistently improves QA, routing accuracy, and cost efficiency across models and both in- and out-of-distribution benchmarks.
>
> Overall, Pre-Route is **not just prompt engineering**, but a **systematic framework for exposing and learning LLMs’ meta-level routing decisions**, with both conceptual and practical impact.

---

> ### Author Response · Authors · 2025-11-17
> **Response to Weakness 3 & 4**
>
> ### W3 Response
> We thank the reviewer for raising this point, but we believe there is a misunderstanding regarding the cost model. Pre-Route is designed with a different objective from modern agentic-RAG or RL-based integrated routing frameworks, and the two are not in conflict.
>
> #### (a) On computational overhead.
>
> Section 3.3 already formalizes our cost model; here we briefly summarize the key conclusions relevant to the concern about extra overhead:
>
> - We decompose total cost into **routing cost $C_\text{route}$** and **answering cost $C_\text{answer}$**. As shown in Table 2, with the distilled router (Qwen3-1.7B), **Pre-Route’s routing cost is about one-fifth of Self-Route**, i.e., it is already cheaper than letting a large model self-route. Section 3.3.3 further shows that **even with a 235B router, the planning cost is much smaller than a single LC pass**, and with a 1.7B router it is only around **1% of a typical LC computation**.
> - More importantly, Section 3.3.2 shows that the **dominant cost comes from answering**, especially from how often LC is invoked: each LC call processes the entire document and is much more expensive than RAG. Thus, reducing $p(\text{LC})$ is the most effective way to reduce total cost.
>
> Pre-Route does exactly this: with a very lightweight routing step—**cheaper than Self-Route itself**—it **significantly reduces LC usage while improving QA quality**. In this sense, Pre-Route is **not “adding another expensive model,” but reducing the dominant answering cost via an almost-negligible routing overhead**, leading to lower end-to-end compute than existing baselines.
>
> #### (b) Distinction from the Concept of Agentic RAG
>
> As discussed in the Introduction and Related Work, our work focuses on the **RAG vs. LC input-choice problem**, which is a different scope from the broader class of agentic-RAG methods. That said, we appreciate the reviewer raising this connection, and we clarify the distinction here. Recent surveys define *agentic RAG* as pipelines with LLMs capable of **reflection, planning, and tool use**, allowing the model to **iteratively decide when/what to retrieve** within a RAG workflow. These methods—such as ReAct, Self-RAG, or Search-R1—optimize **policies over retrieval actions**. From a paradigm perspective, these methods primarily focus on optimizing retrieval-action policies within the RAG paradigm, rather than explicitly deciding between RAG and LC.
>
> Pre-Route targets a different decision layer:
>
> > Given a query and candidate document(s), **choose once between RAG (snippet retrieval) and LC (full-document reading)** before answering.
>
> This routing is **instance-level, single-step, and low-cost**, and directly targets the **input paradigm selection** rather than iterative retrieval control. In this sense, agentic RAG and Pre-Route address different aspects of the problem,
>
> ### W4 Response
> **(1) Effect of the evaluation change.**
>  As we discuss in the paper, LongBench-v2 differs from LaRA not only in data distribution but also in task format and scoring: LaRA uses open-ended QA with a 0–4 graded score that distinguishes “partially” vs. “fully” correct answers, while LongBench-v2 reformulates tasks as 4-way MCQ with binary accuracy. Under this binary metric, partially correct or vague answers from Self-Route are no longer penalized, so its scores can appear higher even though the underlying answer quality has not actually improved.
>
> **(2) Generalization is still strong.**
> Pre-Route continues to generalize well on LongBench-v2: it reduces LC usage by about 10%, further lowers total cost after distillation, and yields around 5% relative gains in routing accuracy, with 1–2 absolute QA point improvements in most settings. It also behaves consistently across different retrieval configurations, indicating robust performance advantages despite the change in task format.

---

> ### Author Response · Authors · 2025-11-17
> **Response to Question 1-3**
>
> ### Q1 Response
> We thank the reviewer for raising these questions. The linear-probing setup **matches the main experimental configuration**. Key settings are:
>
> **(a) Training size & (b) Data split**
>
> - Distilled Pre-Route uses the same **70/10/20 LaRA split** as the main experiments and is evaluated on the **same test set** (Table 10 distribution).
> - The linear probe is fitted on frozen penultimate-layer representations from the LaRA test split. Since the probe is only used for diagnostic analysis of representation separability—and the backbone LLM is frozen—this does not induce any data leakage or affect the main model training or evaluation.
>
> **(c) Optimization**
>
> - **AdamW**, learning rate **2e-5**, **6 epochs** with early stopping (same SFT configuration as the main results).
>
> **(d) Label construction**
>
> - **Ideal labels:** Eq. (3) in Sec. 3.4 (RAG-vs-LC based on true QA scores).
> - **Route labels:** teacher model decisions.
> - **Doc/Task labels:** directly from LaRA metadata.
>
> ### Q2 Response
>
> Thank you for the reviewer’s careful reading of Table 2 and the targeted questions. We provide a clearer explanation below.
>
> **(a) Experimental configuration and averaging scheme**
>
> All cost metrics in Table 2 (including input/output token counts and inference cost) are computed as **averages over the LaRA benchmark**. For each routing method, we compute the average per-decision routing cost over the full test set, to ensure that the statistics are representative, rather than based on a small number of samples or “representative” tasks. We have further clarified this point in the revised version.
>
> **(b) On the similar token counts of the two models in the Pre-Route stage**
>
> The reviewer pointed out that Qwen3-235B and Qwen3-1.7B have very similar, output token counts in the Pre-Route stage, which can indeed be confusing. To carefully verify this concern, we reran the qwen3-1.7B Pre-Route procedure on the LaRA test set and updated the corresponding entries in Table 2 accordingly.
>
> The re-computed results show that the input/output token counts of the two models in the Pre-Route stage are **indeed very close**. This is mainly because: They use **exactly the same prompt template and routing instruction format**; The lengths of the generated reasoning chains are also broadly similar in practice.
>
> Based on this, we made a small update **only to the output token count entry** in the table, so as to more accurately reflect the recomputed average. It is important to emphasize that this update leads to only a very minor adjustment in that token-count entry; **all other numbers in Table 2 (including all cost values) remain unchanged, and none of the analyses or conclusions in the paper related to cost are affected.**
>
> **(c) Why Self-Route has more input tokens than Pre-Route**
>
> As the reviewer correctly observed, Self-Route has significantly more input tokens than Pre-Route (2600 vs. 1205). This is because Self-Route includes additional retrieved chunks in the routing decision, whereas Pre-Route relies only on metadata (such as document title, type, length, and the beginning of the document) for reasoning.
>
> This design difference directly reflects the key cost contrast between the two methods: Pre-Route performs lightweight “pre-routing” using only low-cost metadata, while Self-Route already accesses retrieved content during the routing stage, which leads to a higher input burden.
>
> ### Q3 Response
> We thank the reviewer for pointing out the **3.51 QA score** of *Always-LC (Baseline)* under the **Qwen3-235B [T]** configuration in Table 3.
>
> This value is **not an error**. *Always-LC* is a strategy that **always uses the full long context for every query**, so it can indeed reach relatively high QA scores in some settings. As clearly stated in Section 4.2 and in the table caption, however, *Always-LC* is included **only as an upper-bound baseline in terms of computation**. Because it calls LC on every example, its cost is prohibitively high, so it is **excluded from the main QA ranking and fair performance comparison**.
>
> The purpose of Table 3 is to compare methods under a **controlled computation budget (LC usage < 100%)**. In this regime, Pre-Route achieves QA scores close to (or even better than) Always-LC **while using LC much less frequently**, which is exactly the efficiency benefit we aim to highlight.

---

> ### Author Response · Authors · 2025-11-26
>
> Dear Reviewer 5nQ3,
>
> Thank you once again for your constructive comments on our submission.
>
> As the discussion phase is coming to a close, we would like to respectfully confirm whether our detailed response and the additional experiments have sufficiently addressed your concerns.
>
> We hope that our revisions have resolved the issues you raised. If you are satisfied with our responses, we would greatly appreciate your consideration in adjusting the evaluation score accordingly.
>
> Thank you again for your time and engagement.
>
> Best regards, The Authors

---

> > ### Comment · Reviewer_5nQ3 · 2025-11-26
> >
> > Thanks for the author's detailed responses. The authors' reply has addressed some of my concerns, but I believe issues remain regarding innovation and routing costs.
> >
> > The implementation still relies on carefully designed prompt decision rules, which constitute an explicit heuristic decision tree. This structured guidance itself makes the task requirements clearer and improves model performance. The BoN and linear probes experiments seem to further validate "better prompts → better performance," and I don't think the conclusion about discovering model potential is unique. Besides, distillation itself is a mature technique, and using large model outputs as supervision signals to train small models is not innovative. Moreover, method effectiveness and good experimental results do not equate to method innovation. This work is essentially just a more detailed prompt template plus standard knowledge distillation, lacking innovative design. For a methods paper, this does not meet the innovation standards of a top-tier ML conference like ICLR. Additionally, the paper ignores system-level overhead. The authors only calculate the token cost of a single inference, completely overlooking: the need to deploy an additional 1.7B routing model, latency costs where every request must first call the router adding an extra network round trip, and concurrency bottlenecks where the router may become a system bottleneck in high-QPS scenarios.
> >
> > I have also reviewed the comments from the other reviewers as well as the authors' corresponding replies. I decided to keep my score.

---

> ### Author Response · Authors · 2025-11-27
> **Further Clarifications on System Overhead and Innovation**
>
> Dear Reviewer 5nQ3,
>
> Thank you for taking the time to review our paper and for your detailed comments. Your concerns regarding **system overhead** and **innovation** are well-taken. They have helped us critically re-examine how we present our contributions. We have carefully considered your points and offer the following clarifications.
>
> **1. On System Overhead: Why Pre-Route Reduces Total Load**
>
> We understand your concern that adding a 1.7B router might look like an extra burden. However, Pre-Route is explicitly designed to **reduce the system load** by **using a low-cost module to optimize high-cost operations**.
>
> - **Substantial Cost Reduction:**
>
> As shown in **Table 2**, Pre-Route reduces the per-decision routing cost to **0.16×10⁻³ USD**—only **1/5th** of the Self-Route baseline. The logic is simple: we replace "expensive computation" (using the large model for routing) with "cheap computation" (the 1.7B router). **More importantly, this significantly cuts down calls to the **Long-Context (LC)**, which is the dominant source of cost.** The savings from avoiding expensive LC calls far outweigh the negligible cost of the router.
> - **Alleviating Bottlenecks:** A Qwen3-1.7B router has **significantly higher throughput** than a Long Context Qwen3-235B generator. In any pipeline, the bottleneck is the slow generation of the large model, not the fast classification of the router. By filtering tasks so that the **LC part only handles what is strictly necessary**, Pre-Route actually reduces queuing latency and relieves overall concurrency pressure.
>
> **2. On Innovation: Systematizing Latent Capabilities** We agree that complexity does not equal innovation. However, we believe innovation also lies in finding simple, effective frameworks to solve complex problems.
>
> - **Making Latent Capabilities Explicit:** Our core contribution is discovering that pretrained LLMs already possess the latent knowledge to judge "RAG vs. LC" based on metadata. As our **linear probing (Table 1)** shows, our method doesn't just apply heuristics; it makes the model's representation space linearly separable. This proves the decision is based on the **model's own awareness of its capability boundaries**, not hard-coded rules.
> - **Solving a Real-World Problem:** Pre-Route is not just a prompt template; it is a framework that **factorizes the RAG-LC trade-off** into explicit dimensions. **Pre-Route establishes a routing paradigm where a small model utilizes metadata to perceive the capability boundaries of a large model. This effectively addresses the widely discussed RAG vs. LC dilemma. We believe this represents a non-trivial scientific insight, offering the community a scalable technical path alongside tangible practical value for cost reduction and improving RAG vs. LC practical performance.**
>
> **Conclusion** We genuinely appreciate your detailed review. These discussions have helped us sharpen our arguments regarding the design logic and research value. We hope this response clears up the issues you raised. **If you find that these clarifications have effectively addressed your concerns, we would be grateful if you could improve your score.**
>
> Best regards, The Authors

---

### Official Review · Reviewer_LR4T · 2025-10-31

**Soundness:** 2
**Presentation:** 3
**Contribution:** 3
**Rating:** 6
**Confidence:** 4

**Summary:**

This paper introduces Pre-Route, a proactive framework to address the challenge of choosing between efficient Retrieval-Augmented Generation and costly Long-Context methods. Unlike reactive models, Pre-Route uses lightweight metadata for structured reasoning before answering, enabling it to make explainable and cost-efficient routing decisions. The study shows this approach activates LLMs' latent routing abilities and can be distilled into smaller models.

**Strengths:**

1. This paper introduces "Pre-Route," a novel framework that shifts from a reactive to a proactive "plan-then-execute" paradigm. By performing structured reasoning before generating an answer, it makes more intelligent, cost-efficient decisions and outperforms existing baselines.
2. The research empirically proves that LLMs have a latent, inherent ability for routing.
3. The complex routing logic can be successfully distilled into smaller, lightweight models. This transforms the sophisticated reasoning into a practical, plug-and-play module, enabling efficient and low-cost deployment in real-world applications where large models are not feasible.

**Weaknesses:**

1. The experimental setup for Figure 2 is highly unclear, as key details such as the task type and context length are not specified. Furthermore, it is not explained how the "answer directly" and "unconstrained CoT" methods were implemented. The authors also fail to explain why Pre-route's performance is even inferior to the other two approaches when the Best-of-N sample size is 4 or greater.
2. The main experiments lack a direct comparison against the "answer directly" and "unconstrained CoT" settings.
3. The scope of Pre-Route is presented as overly constrained, limited to the binary choice between RAG and LC. Nevertheless, considering the described methodology and experimental analysis, this framework shows potential for application in a broader spectrum of model routing scenarios.
4. The tables on pages 8 and 9 are difficult to read because the text is too small and the layout is disorganized. The authors should select and present only the most critical information in the main text and move the rest to the appendix.

**Questions:**

please see Weaknesses

---

> ### Author Response · Authors · 2025-11-17
> **Response to Weakness 1**
>
> We thank the reviewer for the thoughtful comments. To make the setup in section 2 clearer, we include additional details in the revision.  We address your three subpoints in turn:
>
>
>
> **(1) Task types and context-length settings**
>
> The experiments in Figure 2 use exactly the same setup as the main experiments and are all conducted on the **LaRA test split**:
>
> - **Task types:** identical to those in Table 10 (location, reasoning, comparison, hallucination).
> - **Context lengths:** we use the original LaRA document lengths (32k/128k), without truncation.
>
> We will add this clarification to the corresponding section in the revised manuscript.
>
>
>
> **(2) How “Answer Directly” and “Unconstrained CoT” are implemented**
>
> As we explained in Section 2.1, all three curves (Pre-Route / Answer Directly / Unconstrained CoT) use **exactly the same metadata inputs**. The only difference lies in the routing instruction format:
>
> All three methods (Pre-Route / Answer Directly / Unconstrained CoT) use the **same models, data, and metadata**.
>  The only difference is the prompt format:
>
> - **Pre-Route:** uses the 6-step structured reasoning described in the paper.
> - **Answer Directly:** same metadata, but the model is asked to **directly choose RAG or LC**, without structured steps.
> - **Unconstrained CoT:** same metadata, but the model is allowed **free-form thinking**, with no imposed structure.
>
>
>
> **(3) Why does Pre-Route appear less advantageous when Best-of-N is large?**
>
> The reviewer correctly observes that when **Best-of-N ≥ 4**, the curves may suggest that Pre-Route improves less steeply than the other two methods. This can be misleading without the following context:
>
> 1. **Starting point & performance ceiling: Pre-Route’s single sample is already near the ceiling.**
>
>    - At **N = 1**, Pre-Route already achieves accuracy close to the “performance ceiling” (estimated using large-N sampling).
>    - In contrast, Answer Directly and Unconstrained CoT perform substantially worse at N = 1, indicating that their **single-sample decisions are noisier and less stable**.
>
>    As a result:
>
>    - Pre-Route’s accuracy gains saturate early because **a single sample is already near a good solution**.
>    - Unconstrained CoT shows a much steeper improvement because Best-of-N essentially performs **multiple trials in a noisy decision space**, increasing the chance of sampling at least one good reasoning path.
>
>    In other words, the steep CoT curve reflects **BoN compensating for unstable single-sample predictions**, whereas the flat Pre-Route curve reflects **already-stable decisions near the ceiling**.
>
> 2. **When considering the cost–performance trade-off, Pre-Route does not actually fall behind.**
>    In the region **N ≥ 4**:
>
>    - The QA performance differences among the three methods become very small.
>    - However, at the same N, **Pre-Route consistently uses much less LC and has lower overall compute cost** than either Answer Directly or Unconstrained CoT.
>
> Thus, under a unified accuracy-plus-cost perspective, Pre-Route does **not** have a real disadvantage. Rather, it reaches near-optimal performance with a single sample and achieves comparable or better QA scores at significantly lower cost when N is large.

---

> ### Author Response · Authors · 2025-11-17
> **Response to Weakness 2–4**
>
> ### W2 Response
> We thank the reviewer for pointing out the lack of direct comparisons with “Answer Directly” and “Unconstrained CoT.” We provide additional clarification below:
>
> **(1) Why these two settings were not listed as separate baselines in the main tables**
>
> The “Answer Directly” and “Unconstrained CoT” settings in Section 2 are already evaluated under the **same setup as the main experiments**. They were introduced to analyze how to elicit routing abilities and ultimately led to our Pre-Route method. In other words, they are **intermediate configurations generated during the development of our method**, rather than full-fledged routing approaches we intend to promote.
>  In the main results, we prioritize comparisons against established routing baselines such as Self-Route, to situate our method relative to prior work. For that reason, these transitional settings were not included in the main tables.
>
> **(2) Additional comparison results: representative numbers**
>
> To directly address the reviewer’s concern, we evaluated Answer Directly and Unconstrained CoT under the exact same experimental setup as the main results. Below we provide representative numbers for two configurations:
>
> - **Large model: Qwen3-235B [T]**
>
>   - **Answer Directly (Q235B)[T]:**
>     QA = 3.34, LC = 47.7%, Acc = 0.51
>   - **Unconstrained CoT (Q235B)[T]:**
>     QA = 3.29, LC = 52.4%, Acc = 0.51
>   - **Pre-Route (Q235B)[T] (ours):**
>     QA = 3.40 (slightly higher), LC = 18.3% (reduced from ~50% to <20%), Acc = 0.68 (substantial improvement)
>
>   This shows that on Qwen3-235B [T], Pre-Route delivers slightly better QA, reduces LC usage from ~48–52% to 18.3%, and improves Acc from 0.51 to 0.68.
>
> - **Small model: Qwen3-1.7B [T]**
>
>   - **Answer Directly (Q1.7B)[T]:**
>     QA = 2.57, LC = 55.1%, Acc = 0.46
>   - **Unconstrained CoT (Q1.7B)[T]:**
>     QA = 2.61, LC = 39.4%, Acc = 0.56
>   - **Pre-Route (Q1.7B)[T]:**
>     QA = 2.87, LC = 11.1%, Acc = 0.80
>
>   On the small model, Pre-Route achieves substantially higher QA and Acc (Acc improves from 0.46/0.56 to 0.80) while reducing LC from 55.1%/39.4% to 11.1%, giving a much better performance–cost trade-off.
>
> **(3) Impact on the main conclusions**
>
> Therefore, even if “Answer Directly” and “Unconstrained CoT” were treated as full baselines in the main comparison, their overall performance in terms of QA, Acc, and LC would still remain clearly inferior to Pre-Route. Including them would not change the conclusions of the paper; rather, it would further strengthen the claim that Pre-Route achieves a superior performance–cost balance.
>  **In the revised version, we already add a more complete table and discussion in the appendix 6.4 to clearly address this point.**
>
> ### W3 Response
> Thank you for the reviewer’s positive comments regarding the potential applicability of the Pre-Route method.
>  We acknowledge that the current experiments in the paper mainly focus on the **binary decision between RAG and LC**, with the goal of validating the core idea and practical effectiveness of the framework.
>
> As noted in **Section 5 (Future Work)**, the paper already outlines plans to extend the framework to broader scenarios, including:“Extending Pre-Route from binary decisions to open-domain settings, exploring more complex routing mechanisms, and incorporating runtime constraints for adaptive deployment.”
>
> Thus, although the current version shows some practical utility, this work represents an initial validation step rather than a limitation of the method itself, and the extensibility naturally comes from the underlying capabilities of modern large models. The core design of Pre-Route—lightweight routing inference based on metadata—**can in principle be naturally generalized to multi-branch model selection, cross-modal reasoning, or more general agent-style task scheduling scenarios**.
>
> ### W4 Response
> We thank the reviewer for the attention to the readability of the result presentation. The tables on Pages 8–9 are indeed dense, but this was a **deliberate choice for scientific completeness**: our goal is to demonstrate Pre-Route’s broad compatibility across six answer-model sizes and two inference modes, rather than showcasing only a few cherry-picked best configurations.
>
> **Regarding presentation clarity:**
>
>  The table already incorporates a visual-guidance design:
>
> 1. **Red shading** highlights the **recommended configurations** (Pre-Route Large & Distilled).
> 2. **Boldface / underlining** mark the **best and second-best** results in each column.
>
> **How to observe quickly:**
>
> As we already mentioned in our paper. The **majority of best/second-best results fall within the red-shaded region**, which directly illustrates that Pre-Route achieves **consistently strong performance across configurations**, rather than depending on a specific model pair. This is the scientific motivation for retaining the full comparison.
>
> We will consider further visual refinements in future revisions depending on available space.

---

> ### Author Response · Authors · 2025-11-26
>
> Dear Reviewer LR4T,
>
> As we enter the final week of the discussion phase, we kindly ask if our clarifications and the previously added experiments have resolved your questions. We would be happy to address any additional points you may have during the remaining time of the discussion phase.
>
> Thank you again for your time and engagement.
>
> Best regards, The Authors

---

> > ### Comment · Reviewer_LR4T · 2025-11-27
> >
> > Thank the authors for their thorough rebuttal. The detailed response has addressed all my concerns regarding the experiments. However, I remain cautious about the limited application scenarios of the proposed method. I decide to stick with my score.

---

### Official Review · Reviewer_srmW · 2025-11-01

**Soundness:** 2
**Presentation:** 3
**Contribution:** 2
**Rating:** 6
**Confidence:** 3

**Summary:**

This paper proposes the Pre-Route framework: an proactive and efficient routing framework that implements a "plan-first, execute-later" model. It uses structured reasoning to select between RAG (Retrieval-Augmented Generation) and LC (Local Computation) before generating responses.  Through experiments such as Best-of-N sampling and linear probe analysis, it proves that LLMs inherently possess the ability to decide whether to use RAG or LC. This capability, however, remains in a "dormant" state. The structured guidance of Pre-Route can effectively "activate" and stabilize this capability.  It successfully distills this complex routing planning capability from expensive large models to smaller models. This enables smaller models to make high-quality routing decisions, significantly reducing deployment costs.  Whether on in-domain (LaRA) or out-of-domain (LongBench-v2) datasets, the Pre-Route framework outperforms existing baseline methods (e.g., Always-RAG, Always-LC, Self-Route), achieving a better balance between performance and cost.

**Strengths:**

The paper breaks away from the passive "switch-after-failure" paradigm (e.g., Self-Route) and proposes a new paradigm of "Proactive Routing" (Pre-Route). This "plan-first, execute-later" approach is logically more superior and efficient.
The most prominent highlight of Pre-Route is that it only uses "low-cost metadata" (such as document type, length, and opening snippets) for planning. This means the computational cost of the "planning" step is nearly zero. It does not require reading the entire document in advance or running an expensive RAG process first just to make a decision. This is highly appealing in engineering practice.
Additionally, the paper successfully distills this complex "planning capability" into a small model with only 1.7B parameters, while still maintaining excellent performance. This enables the Pre-Route framework to serve as a "lightweight, low-cost, and plug-and-play" module.

**Weaknesses:**

1. The Definition of "Ideal Label" May Be Overly Idealized
The paper defines "Ideal Label" as follows: between RAG and LC, the one with a higher QA score is selected (RAG is chosen in case of a tie). This definition is based on an "after-the-fact" omniscient perspective. In practical applications, however, it is impossible to know in advance which method will yield a higher score.
The Pre-Route model learns to fit this "Ideal Label", but the label itself is generated in a controlled environment where "all results are known". Can this be generalized to new, unseen documents and questions? In other words, has the model learned the "statistical patterns" specific to the LaRA dataset, or truly universal "routing reasoning capabilities"? The paper dedicates significant space in Section 2 to arguing this point (e.g., using linear probes), but "correlation" does not equate to "causality".
2. Dependence on Metadata Is a "Weakness"
Pre-Route relies heavily on low-cost "metadata" (titles, lengths, document types, opening snippets).
What if the quality of the metadata is poor?
For web pages, chat records, or code repositories without a clear "document type" (doc_type), the initial reasoning step of Pre-Route may fail.
If the opening snippet (doc_head) of a document is misleading (e.g., the opening resembles a factual retrieval task, but the core lies in complex reasoning at the end of the document), Pre-Route may be misled into selecting RAG.
The paper's experiments seem to focus on relatively well-structured documents (novels, papers, financial reports). Its robustness on "messier", unstructured real-world data has not been fully verified.

**Questions:**

see weakness

---

> ### Author Response · Authors · 2025-11-17
> **Response to Weakness 1**
>
> Thank you for raising this concern. We explain this as follow:
>
> **1. The ideal label is only used as an offline rejection sampling signal, and is never available at inference.**
>  Although the ideal label is defined using actual QA scores, this is fully aligned with standard supervised learning practice. Such signals exist **only** during training. In our method, the ideal labels behave as **rejection / filtering signals in RFT**: after comparing RAG vs. LC, they simply indicate **which routing decisions are worth learning and which samples should be ignored or down-weighted**, without assuming access to QA scores at deployment. At inference time, the router relies **only on metadata m** (task/document types,  etc.).
>
> Importantly, even **without any distillation**, Table 3 in our paper shows that **Pre-Route (Qwen3-235B) [N/T]**, using only structured prompts, already clearly outperforms Self-Route and other baselines. This suggests that the guidelines mainly **activates routing abilities already latent in the large model**, rather than learning a new skill from the ideal labels. The ideal labels are only used as supervision when **compressing this ability into a small model**, indicating what should or should not be distilled. On LongBench—whose formats and scenarios differ substantially from LaRA—the small model still performs well, partly due to the large model’s inherent ability and partly due to effective knowledge distillation.
>
> **2. Our evidence for generalization is more than linear probes and consists of three complementary parts.**
>
>  (i) On LongBench-v2, where document types, task formats, and evaluation schemes differ markedly from LaRA, Pre-Route maintains high routing accuracy and substantially reduces LC usage, showing that its rules transfer beyond the training distribution rather than memorizing LaRA-specific patterns.
>
>  (ii) Linear probing shows that the guidelines routing signal becomes more linearly separable in the representation space, while simple document/task-type labels remain hard to predict, ruling out crude “type-based” heuristics.
>
>  (iii) Ablation studies show that removing any reasoning step causes consistent and interpretable degradation in QA or excessive LC usage.
>
> Together, these results support that the model learns **transferable, structure-aware routing principles**, not just LaRA-specific statistical patterns.

---

> ### Author Response · Authors · 2025-11-17
> **Response to Weakness 2**
>
> We appreciate the reviewer’s concern. While metadata quality may vary, Pre-Route is not built on the assumption of clean or fully available metadata. It relies on **low-cost structural signals that are naturally present in most real-world systems**, and we provide explicit fallback mechanisms when such information is incomplete.
>
> First, in practical RAG and long-context pipelines, document length and the opening snippet are always available, and items such as titles or document types are typically auto-generated from filenames, URLs, or database fields. Real-world documents—webpages, chat logs, code repos, enterprise files—naturally carry structural cues (titles, timestamps, speaker IDs, file paths). These are *system-level byproducts*, not extra assumptions, making reliance on such metadata a realistic engineering choice rather than a fragile dependency.
>
> Second, even when metadata is noisy (e.g., a misleading doc_head), Pre-Route does **not** hinge on any single feature. It integrates query cues, document structure, and information-distribution reasoning. For example, in the Stella case from the appendix 6.6, despite head snippet looks like a local factual snippet of a story, cues like *“repeatedly”* and *“throughout the story”* lead Pre-Route to identify a globally distributed answer, correctly favoring LC. This shows that doc_head acts only as a soft prior, not a determinant.
>
> Third, for cases where metadata is genuinely missing (e.g., webpages or chat logs without clear doc_type), we use a lightweight **Generated-Meta** fallback: a small model produces a pseudo title/type from the query and doc_head, after which routing proceeds normally. Its overhead is minimal and remains far below Self-Route. Experiments show:
>
> - **Head-only** incurs some degradation but is still consistently stronger than Self-Route;
> - **Generated-Meta** nearly closes the gap to **Full-Meta**;
> - Across settings, both the large and distilled Pre-Route variants maintain higher QA and lower LC usage than Self-Route.
>
> Thus, leveraging low-cost metadata is a deliberate engineering choice, not a weakness of the method. In real scenarios, the document head and length are virtually always available, and most other metadata can be extracted from existing fields or simple preprocessing. For the rare cases where metadata is absent, the Generated-Meta fallback provides a reliable solution backed by empirical results. With these mechanisms, Pre-Route remains robust even on “messier,” less structured real-world data, rather than only on idealized well-formed documents.
>
> We **additionally include this discussion and the corresponding experimental results as a new subsection in Section 4 of the revised manuscript.**
>
>
> | Answer Model | Router Model           | Exp            | QA↑  | LC(%)↓ | Acc↑ |
> | ------------ | ---------------------- | -------------- | ---- | ------ | ---- |
> | DeepSeek-R1  | Pre-Route(Q235B) [N]   | Head-only      | 3.42 | 20.3   | 0.68 |
> |              |                        | Generated-Meta | 3.45 | 20.6   | 0.7  |
> |              |                        | Full-Meta      | 3.47 | 27.2   | 0.7  |
> |              | Pre-Route(D-Q1.7B) [N] | Head-only      | 3.43 | 18.5   | 0.69 |
> |              |                        | Generated-Meta | 3.47 | 15.8   | 0.73 |
> |              |                        | Full-Meta      | 3.51 | 20.6   | 0.73 |
> |              | Self-Route(Baseline)   |                | 3.36 | 31.4   | 0.52 |
> | Qwen3-235B   | Pre-Route(Q235B) [N]   | Head-only      | 3.39 | 23     | 0.65 |
> |              |                        | Generated-Meta | 3.44 | 23.2   | 0.7  |
> |              |                        | Full-Meta      | 3.43 | 27.2   | 0.67 |
> |              | Pre-Route(D-Q1.7B) [N] | Head-only      | 3.4  | 21.5   | 0.66 |
> |              |                        | Generated-Meta | 3.43 | 20.4   | 0.7  |
> |              |                        | Full-Meta      | 3.43 | 22.7   | 0.69 |
> |              | Self-Route(Baseline)   |                | 3.34 | 33.9   | 0.52 |

---

> > ### Author Response · Authors · 2025-11-26
> >
> > Dear Reviewer srmW,
> >
> > With the discussion phase entering its final week, we wanted to follow up to see if our previous response and additional experiments have satisfactorily addressed your concerns. We would be happy to address any additional points you may have during the remaining time of the discussion phase.
> >
> > Thank you again for your time and engagement.
> >
> > Best regards, The Authors

---

### Meta-Review · Area_Chair_GBcg · 2025-12-06

**Summary:**

- The definition of "Ideal Label" may be overly idealized (srmW)
- Dependence on metadata (srmW, bHjd)
- The experimental setup for Figure 2 is unclear (LR4T)
- The main experiments lack a direct comparison against the "answer directly" and "unconstrained CoT" settings (LR4T)
- The scope of Pre-Route is presented as overly constrained (LR4T)
- Lack of justification for design choices (5nQ3)
- Lack of novelty (5nQ3)
- High inference cost and missing comparison with modern agentic RAG paradigm (5nQ3)
- Limited generalization and marginal performance gains (5nQ3)
- Benchmark difficulty (bHjd)

**Reviewer Concerns:**

Most concerns are well addressed but I believe the important concern about novelty is still outstanding.
- Reviewer 5nQ3: "Highly Engineered Approach with Limited Innovation. The method is a highly customized heuristic system—essentially a hand-crafted prompt engineering solution. The six reasoning steps (task & document characterization, distribution pattern judgment, context-window feasibility, retrieval feasibility, model capability consideration, efficiency trade-off) are manually designed heuristics that require pre-specification rather than being automatically learned or generated. This pre-designed nature significantly limits innovation, as the approach lacks unique architectural contributions beyond careful prompt construction. The core claim of "activating latent routing ability" essentially reframes better prompting as a discovery of hidden capabilities, which overstates the conceptual novelty"
  - The rebuttal claims that the main contribution is not the template itself, but the systematic identification and stabilization of a latent RAG–LC routing ability in LLMs. However, the baseline method, Self-Route, does not use such a well-designed prompt template. This paper conducted baseline experiments by using Self-Route's prompt template directly, while the answer model and evaluation datasets are changed. In other words, the claimed novelty of this paper is not well proven by a fair comparison between Pre-Route and an updated version of Self-Route with well-designed prompt template on the target answer models and evaluation datasets. We still cannot draw a conclusion whether the performance gain is from the prompt template or pre-route.

**Reviewer Scores:**

Reviewers srmW & LR4T: keep the positive score of 6.
Reviewer 5nQ3: keep the score of 4 as the key concern about novelty is not well addressed.
Reviewer bHjd would raise the score from 4 to 6 as the concerns are well addressed.

---

### Decision · Program_Chairs · 2026-01-26

Reject